

# IMPROVED ANALYSIS OF SOLAR SIGNALS FOR DIFFERENTIAL REFLECTIVITY MONITORING

Asko Huuskonen[1], Mikko Kurri[1], and Iwan Holleman[2]

[1]Finnish Meteorological Institute, Helsinki, Finland
[2]Radboud University, Faculty of Science, Nijmegen, The Netherlands

*Correspondence to:* Asko Huuskonen (Asko.Huuskonen@fmi.fi)

**Abstract.** The method for the daily monitoring of the differential reflectivity bias for polarimetric weather radars is developed. Improved quality control is applied to the solar signals detected during the operational scanning of the radar which removes efficiently rain and clutter contamination occurring in the solar hits. The simultaneous reflectivity data are used as a proxy to determine which data points are to be removed. A number of analysis methods are compared, and methods based on surface

fitting are found superior to simple averaging. A separate fit to the reflectivity of the horizontal and vertical polarization channels is recommended, because it provides in addition to the differential reflectivity the pointing difference of the polarization channels, the squint angle. Data from the Finnish weather radar network shows that the squint angles are less than 0.02° and that the differential reflectivity bias is stable and determined to better than 0.04 dB. The results are compared to those from measurements at vertical incidence, which allows to determine the differential receiver bias and the transmitter bias.

## 1 Introduction

Calibration of the radar differential reflectivity ($Z_{dr}$) is crucial for the successful use of dual-polarization measurements (Ryzhkov et al., 2005). For example, a bias of only 0.2 dB in the differential reflectivity results in 15% errors on the estimated rain rates (Tabary et al., 2008). Several methods for the calibration exist which make use of either active (transmit and receive) or passive (receive only) observations of unpolarized targets.

The active methods are based on polarimetric properties of rain. Differential reflectivity at vertical incidence is intrinsically zero for raindrops, and hence the measured $Z_{dr}$ (in dB) is an estimate of the system bias (Gorgucci et al., 1999). A full azimuth rotation is used to improve the estimate. The calibration is also doable at oblique angles using observations of light rain in which the rain drops are closely spherical (Cunningham et al., 2013). Methods using rain observations provide calibration of the full transmitter-receiver chain.

The passive methods are based on using the sun. The measurements can be off-line measurements, in which the operational scanning is stopped and the radar antenna is pointing at the sun (Pratte and Ferraro, 1989; Melnikov et al., 2003; Ryzhkov et al., 2005; Zrnić et al., 2006) or they can be on-line measurements, in which data from operational scans is used and the normal radar operations need not be stopped (Holleman et al., 2010a; Figueras i Ventura et al., 2012; Frech, 2013). Unlike rain





calibration, the solar observations provide information only on the receiver chain, but they provide antenna angle information in addition.

Holleman et al. (2010a) introduced the on-line method to the solar $Z_{dr}$ data, and showed daily differential reflectivity biases from French and Danish operational polarimetric radars and compared them to those obtained from rain calibration at zenith. The paper demonstrated the capability and importance of daily monitoring of the $Z_{dr}$. The method builds on those using the solar signals for the antenna pointing, and for the monitoring of the radar receiver chain stability (Darlington et al., 2003; Huuskonen and Hohti, 2004; Holleman and Beekhuis, 2004; Huuskonen and Holleman, 2007; Holleman et al., 2010b). Figueras i Ventura et al. (2012) studied one year of data from French radars and concluded that both the solar method and the zenith calibration fluctuate less than $\pm 0.2$ dB and that the fluctuations stem mainly from the variability of the receiver chain. Frech (2013), by analyzing the data from horizontal and vertical channels separately, was able to determine the squint angle of the antenna finding a value of about $0.02^\circ$. The squint angle results are obtained in addition to results on antenna pointing and receiver power. Cunningham et al. (2013) show results on the operational WSR-88 network. Gabella et al. (2015) describe a simple method for estimating the $Z_{dr}$ bias, which is based on using power differences only.

In the present paper, we revisit the monitoring of the receiver chain of the $Z_{dr}$ calibration by the on-line solar method. We discuss the filtering of the raw solar hit data to remove rain and clutter contamination with aim of increasing the quality and number of the solar hits. We then present several ways to obtain the $Z_{dr}$ bias and discuss their usability. We also study the two polarization data sets separately, which produces results on the pointing difference between the polarizations (squint angle). Results are compared with those obtained from the $Z_{dr}$ calibration in rain with zenith pointing antenna.

## 2 Data

The Finnish Meteorological Institute (FMI) operates a network of ten C-band Doppler weather radars, of which nine radars are polarimetric. Every 15 minutes the radars perform an 11-elevation volume scan between $0.3^\circ$ and $45^\circ$ elevations, where six elevations up to 9 degrees are scanned in single–PRF with 570 Hz, and the others in dual–PRF. At every 5 minutes six of these eleven elevations are repeated. A recent description of the FMI network is given by Saltikoff et al. (2010). The network upgrade to polarimetry has continued since then so that now all radars except one are polarimetric, and two new radars have been added to the network. For convenience some relevant properties of the four radars used in this study are shown in Table 1.

## 3 Method

### 3.1 Detection of sun signatures

The detection of solar signatures in polar volume data of weather radars is described in Huuskonen and Holleman (2007). In the method a reflectivity signal which originates from a continuous microwave source is searched along radials in the operational scan data. Figure 1 shows an example where four radars observe the solar signal simultaneously close to the spring equinox. As a radar signal processor usually corrects the received echoes for the range dependence and the atmospheric attenuation, the



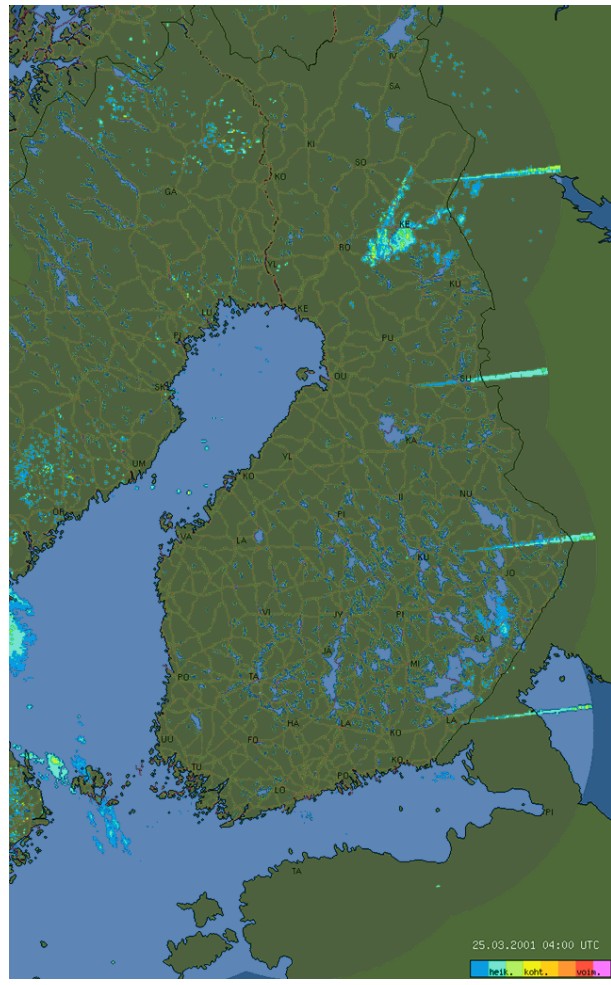

**Figure 1.** Solar signal observed simultaneously by four radars in Finland on March 25, 2001 at 4:00 UTC.

"reflectivity" signals from the sun increase with the range. The received solar spectral power at the antenna feed $P_{H,V}$ for the *H*orizontal and *V*ertical polarizations (per MHz in dBm) can be calculated from the reflectivity signature as a function of the range $Z_{H,V}(r)$ (in dBZ) using:

$$P_{H,V} = Z_{H,V}(r) - 20 \log_{10} r - 2ar - C_{H,V} - 10 \log_{10} \Delta f \qquad (1)$$

5  where $C_{H,V}$ is the radar constant in dB according to Probert-Jones (1962) for horizontal and vertical polarizations, respectively, $a$ the one-way gaseous attenuation in dB km$^{-1}$, and $\Delta f$ the receiver 3 dB bandwidth in MHz, assumed to be the same for both polarization channels. In the case of a proper solar signal the power $P$ is constant along the range. Depending on the hardware of the radar, the volume coverage pattern, the season, and the latitude of the radar, several tens of sun hits are found per day.





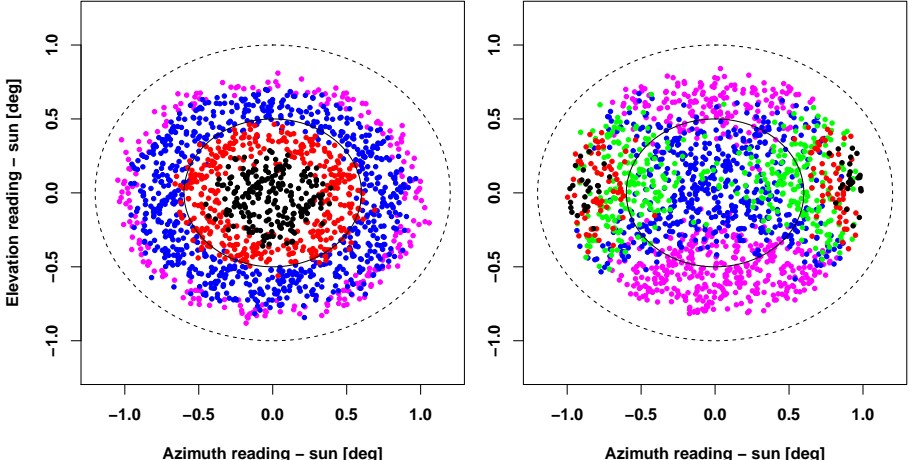

**Figure 2.** Sun images based on sun hits collected from the FMI Anjalankoski radar in March 2015. The left panel shows the sun hit power relative to maximum (black within 1 dB of maximum, red 1 ... 3 dB below the maximum, blue 3 ... 7 dB below the maximum, and magenta more than 7 dB below the maximum) and the right panel the differential reflectivity (magenta less than -0.3 dB, blue -0.3 ... -0.1 dB, green -0.1 ... 0.1 dB, red 0.1 ... 0.3 dB , and black above 0.3 dB). An ellipse with axes of 1º and 1.2º is provided to show the approximate half power (3 dB) widths in elevation and azimuth, respectively. The dashed lines show ellipses with axes lengths twice that.

Uncorrected reflectivity data are used for this analysis as (time-domain) Doppler clutter filters can attenuate the solar signal by several dBs. The solar signal may also have contamination caused by ground clutter and precipitation. This can be circumvented by discarding data below 1º elevation and using data from long ranges (e.g. > 100 km) only.

### 3.2 Modelling of $Z$ signatures

5 The sun hits have a symmetric distribution around the sun position which is a slightly wider in azimuth than in elevation. A typical example is shown in the left panel of Fig. 2. The distribution of linear powers is well approximated by a Gaussian form, and hence the power $P_{H,V}$ for the horizontal and vertical polarizations in decibels can be written as:

$$P_{H,V}(x,y) \equiv a_x \cdot x^2 + a_y \cdot y^2 + b_x \cdot x + b_y \cdot y + c \qquad (2)$$

where the coordinates $x$ and $y$ are defined as:

10 $$x = (\phi_{\text{read}} - \phi_{\text{sun}}) \cdot \cos\theta_{\text{sun}} \qquad (3)$$
$$y = \theta_{\text{read}} - \theta_{\text{sun}} \qquad (4)$$

where $\phi$ and $\theta$ denote azimuth and elevation, "read" refers to the angle reading of the radar antenna and "sun" refers to the calculated sun position. The observed azimuth differences are multiplied by $\cos\theta_{\text{sun}}$ to make them invariant to the elevation





**Table 1.** Some properties of the FMI radars relevant to this study. The columns give the radar name, three letter acronym, latitude and longitude of the radar, the antenna beam widths for the horizontal polarization (H) and vertical polarization (V) in elevation and azimuth directions in degrees, the difference of losses in the transmitter chain ($T_x$) in dB, and difference of the antenna gains ($g$) in dB. The beam widths in the elevation and azimuth directions are based on measurement of the electric (**E**) and magnetic fields (**H**), as indicated in the parenthesis.

| Radar | Code | lat $^\circ$ N | lon $^\circ$ E | H(el,**H**) | H(az,**E**) | V(el,**E**) | V(az,**H**) | $T_{x,h} - T_{x,v}$ | $g_h - g_v$ |
|---|---|---|---|---|---|---|---|---|---|
| Korpo | KOR | 60.13 | 21.64 | 0.914 | 0.980 | 0.940 | 0.941 | 0.0 | -0.1 |
| Anjalankoski | ANJ | 60.90 | 27.11 | 0.909 | 0.974 | 0.967 | 0.923 | 0.0 | 0.0 |
| Kesälahti | KES | 61.91 | 29.80 | 0.927 | 0.949 | 0.944 | 0.928 | 0.0 | -0.2 |
| Kuopio | KUO | 62.86 | 27.38 | 0.916 | 0.944 | 0.948 | 0.908 | -0.1 | 0.1 |
| Luosto | LUO | 67.14 | 26.90 | 0.911 | 0.983 | 0.953 | 0.943 | -0.2 | 0.0 |

(e.g. Doviak and Zrnić, 1993, p. 516). We assume that the biases and the widths are independent of the pointing angles, and that the microwave center of the sun is close to the center of the optical disk. Eq. 2 is linear in the parameters $a_x$ to $c$, and thus the sun data can easily be fitted to this equation by the least squares method. Parameters $a_x$ and $a_y$ are related to the widths of the distributions of the $x$ and $y$ values, parameters $b_x$ and $b_y$ to the elevation and azimuth biases, and parameter $c$ to the

peak power, i.e. when the antenna is pointing exactly at the sun. Note that these parameters are different for the horizontal and vertical polarizations. The elevation width $\Delta_\theta$, the azimuth width $\Delta_\phi$, the elevation bias $B_\theta$, the azimuth bias $B_\phi$, and the power when the antenna is pointing directly to the sun, $\hat{P}_{\mathrm{sun}}$, can be calculated from the linear parameters (Huuskonen and Holleman, 2007):

$$\Delta_{\phi,\theta}^2 = -\frac{40 \log_{10} 2}{a_{x,y}} \approx -\frac{12}{a_{x,y}}, \tag{5}$$

$$B_{\phi,\theta} = -\frac{b_{x,y}}{2a_{x,y}}, \tag{6}$$

$$\hat{P}_{\mathrm{H,V}} = c - \frac{b_x^2}{4a_x} - \frac{b_y^2}{4a_y}. \tag{7}$$

The widths are obtained from Eq. 5 when the corresponding parameter $a_{x,y}$ is negative.

As data from different elevations are analyzed together the solar elevation needs to be corrected for the effects of refraction. We use the analytical formulas for the atmospheric refraction of radiowaves, which are consistent with the commonly used

$k$-model or 4/3-model (Holleman and Huuskonen, 2013). As the solar signal traverses all atmosphere, the expected value of $k$ is less than 4/3, which is valid close to the surface. Hence we use $k = 5/4$, which fits best to the model calculations and radar observations according to Holleman and Huuskonen (2013). To avoid severe refraction, data below 1$^\circ$ elevation are discarded.





### 3.3 Modelling of $Z_{\mathrm{dr}}$ signatures

The right panel of Fig. 2 shows a scatter plot of the differential reflectivity, based on the same data as the reflectivity in the left panel. The distribution looks very different from that of the reflectivity, as the curvature has opposite signs in the elevation and azimuth directions. The distribution has a form of a saddle surface. In the vertical (elevation) axis the values are negative at the

5    edges indicating that the V lobe is wider than the the H lobe, whereas in the azimuth direction the opposite is true, in qualitative agreement with the antenna measurements shown in Table 1. The saddle surface form is different from the symmetric form seen in the Trappes radar data (Holleman et al., 2010b).

We can determine $Z_{\mathrm{dr}}$ at the direct solar pointing using the same formulation as used for the reflectivity. We hence write $Z_{\mathrm{dr}}$ as:

$$Z_{\mathrm{dr}} \equiv \bar{a}_x \cdot x^2 + \bar{a}_y \cdot y^2 + \bar{b}_x \cdot x + \bar{b}_y \cdot y + \bar{c} \tag{8}$$

where $x$ and $y$ are related to azimuth and elevation as given by Eq. 3 and Eq. 4. Noting that the curvature may be of either sign, the elevation width $\bar{\Delta}_\theta$, the azimuth width $\bar{\Delta}_\phi$, the elevation bias $\bar{B}_\theta$, the azimuth bias $\bar{B}_\phi$, and the $Z_{\mathrm{dr}}$ when the antenna is pointing directly to the sun, $\hat{Z}_{\mathrm{dr}}$, can be calculated from the linear parameters as

$$\bar{\Delta}_{\phi,\theta}^2 = \frac{40 \log_{10} 2}{|\bar{a}_{x,y}|} \approx \frac{12}{|\bar{a}_{x,y}|}, \tag{9}$$

$$\bar{B}_{\phi,\theta} = -\frac{\bar{b}_{x,y}}{2\bar{a}_{x,y}}, \tag{10}$$

$$\hat{Z}_{\mathrm{dr}} = \bar{c} - \frac{\bar{b}_x^2}{4\bar{a}_x} - \frac{\bar{b}_y^2}{4\bar{a}_y}. \tag{11}$$

where $|\bar{a}_{x,y}|$ is introduced to obtain the width independent of the sign of the curvature parameter.

If the horizontally and vertically polarized beam patterns of the radar antenna were fully identical, the curvature parameters $\bar{a}_x$ and $\bar{a}_y$ would be zero, the received differential reflectivity constant, and simple averaging of the $Z_{\mathrm{dr}}$ would be sufficient.

20    In all other cases, the curvature provides information on the difference between the widths of the horizontal and vertical polarization beams. Following the procedure of Holleman et al. (2010a), we can derive a relation between the widths by noting that

$$Z_{\mathrm{dr}} \equiv Z_H - Z_V = P_H - P_V + (C_H - C_V) \tag{12}$$

where we have applied Eq. 1. The difference of the radar constants does not affect the form of the distribution. Hence we can

25    expand the powers with Eq. 2. By equalling the coefficients of the two-dimensional polynomial, we arrive at the following



equations:

$$\frac{1}{\overline{\Delta}_\phi^2} = \frac{1}{\Delta_{\phi,v}^2} - \frac{1}{\Delta_{\phi,h}^2} = \frac{\bar{a}_x}{40\log_{10}2} \tag{13}$$

$$\frac{1}{\overline{\Delta}_\theta^2} = \frac{1}{\Delta_{\theta,v}^2} - \frac{1}{\Delta_{\theta,h}^2} = \frac{\bar{a}_y}{40\log_{10}2} \tag{14}$$

$$\bar{B}_{\phi,\theta} = -\frac{\bar{b}_{x,y}}{2\bar{a}_{x,y}}, \tag{15}$$

$$\hat{Z}_{\mathrm{dr}} = \hat{P}_H - \hat{P}_V + (C_H - C_V) = \bar{c} - \frac{\bar{b}_x^2}{4\bar{a}_x} - \frac{\bar{b}_y^2}{4\bar{a}_y} \tag{16}$$

Many radar software systems do not provide the two reflectivities, but instead the horizontal reflectivity and the differential reflectivity (which is often calculated by subtracting detected powers and not reflectivities). Then one can derive the vertical reflectivity as $Z_V = Z_H - Z_{\mathrm{dr}}$. This approach implies that $C_H = C_V$ in Eq. 12.

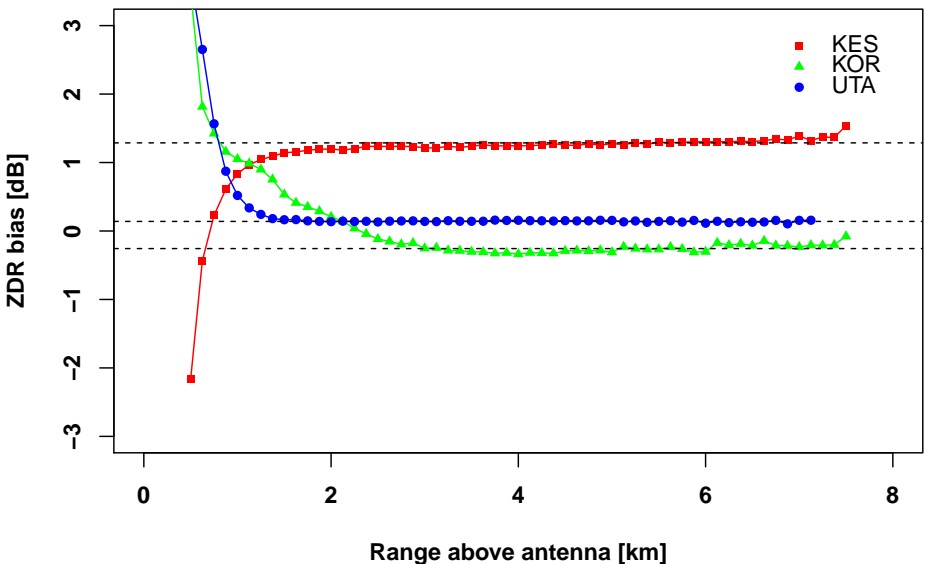

**Figure 3.** $Z_{\mathrm{dr}}$ as a function of altitude above the antenna estimated from zenith measurements during rain events. Each point is a daily average of measurements for a full azimuth rotation of the antenna. The dashed lines indicate the $Z_{\mathrm{dr}}$ bias calculated as explained in the text.

### 3.4 Calibration of $Z_{\mathrm{dr}}$ during rain

Calibration of differential reflectivity using polarimetric properties of rain has first been demonstrated by Gorgucci et al. (1999). Differential reflectivity at vertical incidence is intrinsically zero for raindrops, and hence the measured $Z_{\mathrm{dr}}$ (in dB)





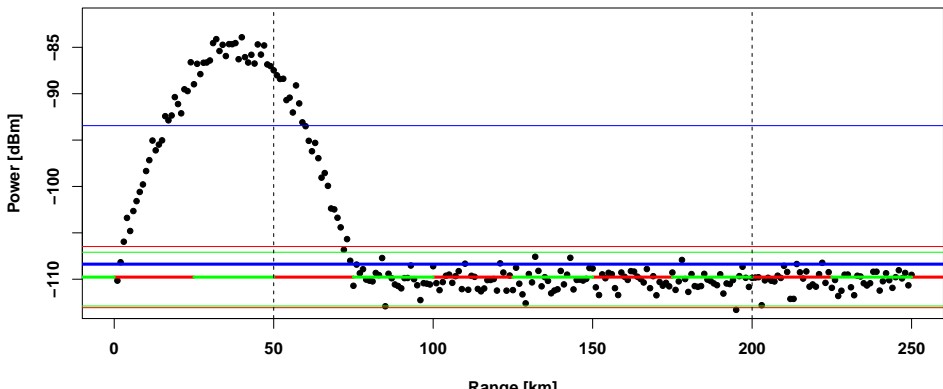

**Figure 4.** Statistical estimators in simulated rain with noise. The thick lines give three estimators of the center point, mean from 50 km distance (blue), median from 50 km distance (red), mean from 200 km distance (green). The thin lines of the respective color indicate the filtering windows. The second blue thin line at -123 dB is not shown. The window width for the mean estimator is three times the standard deviation, and for the median estimator is three times the median absolute deviation scaled by 1.4280 so that it agrees with the standard deviation of the underlying normal distribution. The the two range limits are indicated be vertical dashed lines.

is an estimate of the system bias. The differential reflectivity is interpreted as the sum of the differential biases in reception, $\Delta R_{\mathrm{dr}}$, and transmission, $\Delta T_{\mathrm{dr}}$:

$$Z_{\mathrm{dr}}^{\mathrm{rain}} = \Delta R_{\mathrm{dr}} + \Delta T_{\mathrm{dr}} \tag{17}$$

where $\Delta R_{\mathrm{dr}}$ and $\Delta T_{\mathrm{dr}}$ consist of differences of waveguide and other losses and the antenna gain between the horizontal

and vertical polarization channels, and $\Delta T_{\mathrm{dr}}$ is in addition affected by the differences in the transmitted power between the channels.

For $Z_{\mathrm{dr}}$ bias estimation, all FMI polarimetric radars scan 360 degrees in azimuth with vertically pointing antenna every 15 minutes. There is a transient feature during the first few kilometers from the radar, because the horizontal and vertical receiver channels return to their normal operational gain in a different way after the transmission pulse. Range of this effect is radar

dependent and varies from 2 km to 8 km. Because of transient effect, $Z_{\mathrm{dr}}$ bias is analyzed from altitudes where the transient has died out. Figure 3 shows an example of the $Z_{\mathrm{dr}}$ bias analysis. In the analysis an average of $Z_{\mathrm{dr}}$ values of the sweep over 360 degrees in azimuth is calculated for each altitude level defined by the 125 m range gate used in these measurements. As a quality measure, only data with cross correlation coefficient $\rho_{HV} > 0.9$ and SNR $> 5$ dB are accepted for the analysis, and data beyond two standard deviations from the mean are discarded. If more than 80 % of the 360 degree azimuth sector is covered

by good quality data points the mean $Z_{\mathrm{dr}}$ bias is calculated for that altitude level. This last requirement is used to remove the potential $Z_{\mathrm{dr}}$ bias caused by the orientation symmetry of the scatterers or by other disturbances during the measurement.





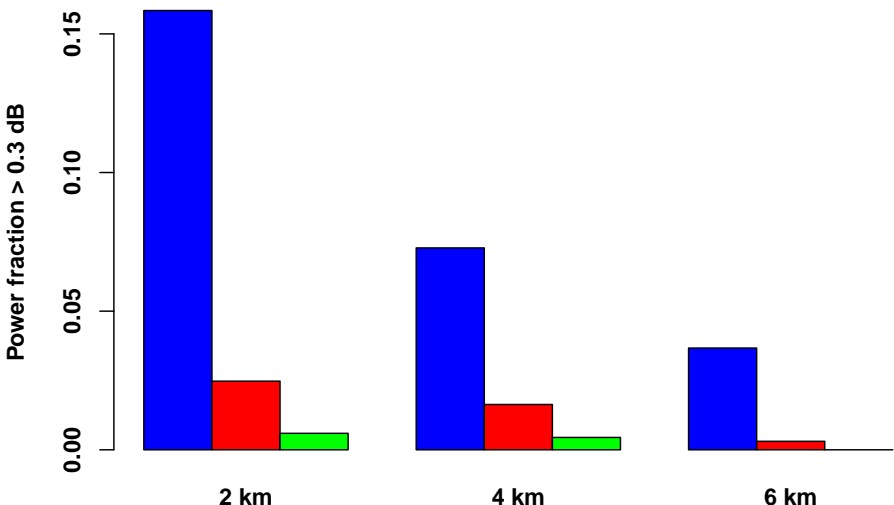

**Figure 5.** Probability that the solar hit power exceeds the far range power by 0.3 dB. The power determined by mean filtering is denoted by blue, and the power determined by median filtering by red. The green bar indicates solar power determined by the mean filtering when the filtering window is determined from points above 8 km altitude. The probability is given for start altitudes of 2, 4, and 6 km.

This procedure is repeated for every vertical scan and the final result is calculated as a daily mean of these quality controlled measurements.

## 4 Results and discussion

### 4.1 Quality control of the solar hits

5 The quality control of the solar hit data is necessary to get results of good quality. Huuskonen and Holleman (2007), Holleman et al. (2010b) and Huuskonen et al. (2014a) used a two stage approach for ensuring that rain and clutter do not contaminate the hit data. The first stage is to use data from far ranges only (i.e. 100 km (Huuskonen et al., 2014a)) which guarantees that clutter is not included, and the hit average is made of points above the rain in most cases. In addition solar hits with standard deviation much larger than that expected for a solar hit (e.g. three times larger) have been rejected. The second stage was to 10 do the fitting of data to the model twice. After the first fit, data points too far (e.g. 1 dB) from the fitted curve are removed and a second fit is carried out on the remaining points. This method works efficiently when a small number of outliers appears in the data. The results are further improved if one assumes that the antenna is already well pointed and uses only hits close





to that direction. This is an efficient method to avoid using signal from RLANs (Radio Local Area Network) which produce signatures resembling those of the sun. The use of these methods guarantees that good results are obtained in a majority of cases. As the antenna pointing or the calibration levels are not adjusted on a daily basis based on these results, occasional bad results cause no trouble.Altube et al. (2015) describe a partially similar set of methods which offer the same functionality for

the improvements of the quality of solar hits.

The number of sun hits and also the statistical accuracy can be increased by using data also from closer ranges. Then the probability that the data is contaminated by rain or clutter increases, and one has to device a method of removing the contaminated data prior to calculating the solar hit power. One possible method is a two-stage estimation, in which the first estimate of the solar hit power is calculated from the full dataset, and this power with an estimate of the width of the distribution

is used to remove non-solar data. This is illustrated in Fig. 4 which shows a simulated solar hit case with rain at range less than 70 km. The figure shows that the mean of data between the 50 km and the 250 km ranges (blue) is much higher than the median of the same data (red), which in turn is close to the mean of data beyond the 200 km range (green). If the last is taken as an unbiased estimator of the solar power, the conclusion is that the median is a much better estimator than the mean, when all data beyond 50 km is used. The median represents the solar power as long as clearly more than half of the points

are genuine solar hit points, but will of course be more biased when the amount of contaminated points increases. Evidently, the power estimate based on far ranges only is even better. The estimates of the width of the distribution confirm the above. If the standard deviation is used to determine the width of the distribution, the estimate is biased by the solar contamination, as indicated in the figure. The median absolute deviation (MAD) gives a width estimate which is close to width estimated from far ranges only.

Figure 5 shows the result of applying the methods to one month of solar hit data from the Anjalankoski radar. Instead of fixed range limits, as in Fig. 4, altitude limits are used, which enables us to use data from several elevation angles together. The figure illustrates the probability that any of the estimators produce a biased estimate of the solar hit power. The reference power is obtained by calculating the mean hit power above 8 km, which is free of rain contamination for the data used in the study. Figure 5 shows that filtering by mean produces biased estimates even if the start altitude is put to 6 km, and that the number of

biased estimates is significant for lower start altitudes.

The overall best performance is obtained if the filtering window is first estimated using data from high altitudes. Then a biased estimate is obtained only in very few cases. The median estimator, although nearly as good, produces biased estimates of the power in some cases. One can reduce the computational load for real-time analysis of large data sets, e.g. for the analysis of all European solar hit data within the OPERA data center (Huuskonen et al., 2014b) by fixing the width of the filtering

window instead of estimating it for each solar hit. This is possible because the statistics of the solar hit data do not vary from case to case, as pointed out by Altube et al. (2015).

The filtering of the differential reflectivity $Z_{\mathrm{dr}}$ is not as straightforward as the filtering of the reflectivity, because $Z_{\mathrm{dr}}$ in rain and from the sun do not deviate significantly from each other. The data contaminated by rain can be removed prior to averaging by using the reflectivity data as a proxy to indicate which data points are based on sun and which on rain. Hence the estimates



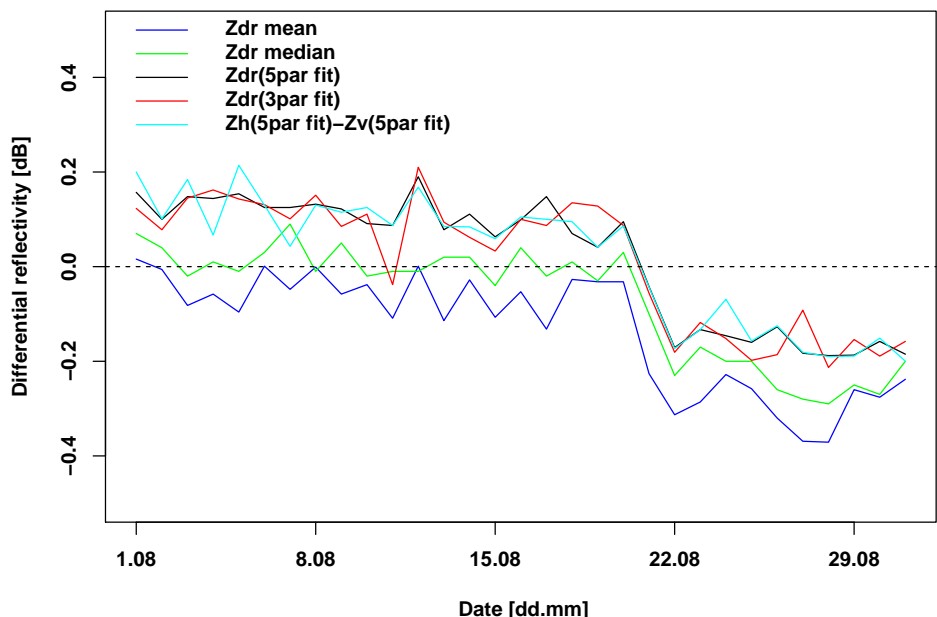

**Figure 6.** $Z_{dr}$ bias values for the ANJ radar in August 2014. The methods are indicated in the figure. The downward step on August 21 is a result of the change of the system $Z_{dr}$ bias.

for $Z_{dr}$ are calculated as the mean and standard deviation of the $Z_{dr}$ profile by applying the same range indices as had been used for the reflectivity data.

## 4.2 $Z_{dr}$ results based on solar hits

There is a number of different methods to estimate the $Z_{dr}$ bias from the solar hits. Holleman et al. (2010a), when solving

5 Eg. 2 for the $Z_{dr}$, first estimated the widths using a larger dataset and fixed the widths in the fitting, to improve the stability of the fit. This is one of the several methods available for the estimation of the $Z_{dr}$ bias. An obvious second choice is to do a full 5-parameter fit, such as recommended for reflectivity data (Huuskonen et al., 2014a; Altube et al., 2015). It is also possible to do 3-parameter fitting by fixing the pointing to that obtained from the reflectivity fit, and thus fitting for power and the two width values only. Noting that the $Z_{dr}$ bias would be constant over the solar disk for fully matching antenna beams, one can

10 take a mean or a median of all $Z_{dr}$ hits. And, finally, it is possible to analyze the reflectivity channels separately, either by a 5- or 3-parameter fit, and get the estimate by differencing the power.

Figure 6 compares $Z_{dr}$ estimates by five methods presented above for one month of data. One can readily notice that the results obtained by using the mean or the median are clearly different from those by the fitting methods. This is not surprising,





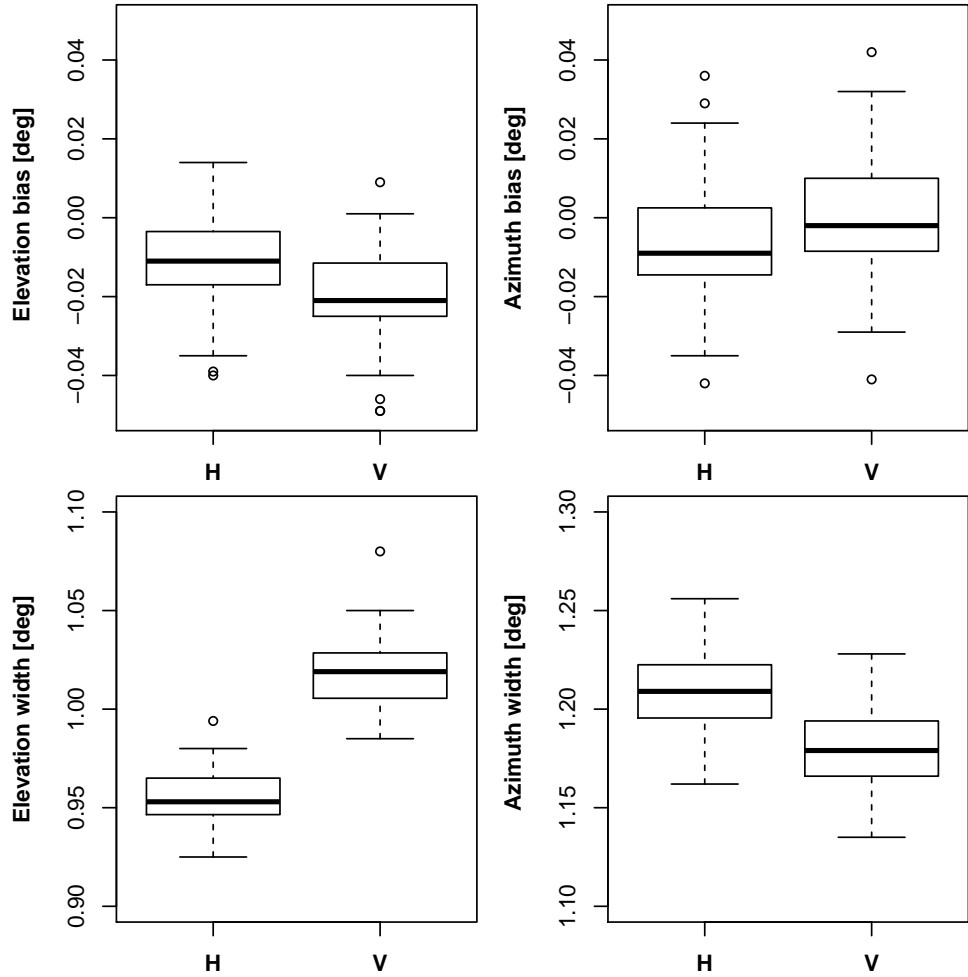

**Figure 7.** Elevation and azimuth pointing and image width results for the H and V polarization data for the ANJ radar. In each panel, the median is shown with a thick line, at 1st and 3rd quartiles by a box. Whiskers indicate data points closer than 1.5 times the box length from the quartiles, and data points beyond that are marked with circles.

because the $Z_{dr}$ field seen in Fig. 2 is not constant. Taking a mean might work for a saddle surface if the averaging is restricted to a small area around the solar direction, but would not be a good method even in that case if the $Z_{dr}$ surface has a clear minimum or maximum at the solar direction, such as for the case shown in Holleman et al. (2010a). Hence these methods are not recommendable. The three fitting methods give comparable results in the FMI case, where the widths of the two

5  polarizations differ, and hence the surface is not close to constant. For surfaces approaching constant the fitting to $Z_{dr}$ may become increasingly unstable, especially if the pointing is one of the fit parameters. Hence a separate 5-par fit to both $Z_H$ and $Z_V$ is the safest method, and hence recommended. The $Z_{dr}$ result is then obtained as the difference of the determined powers.





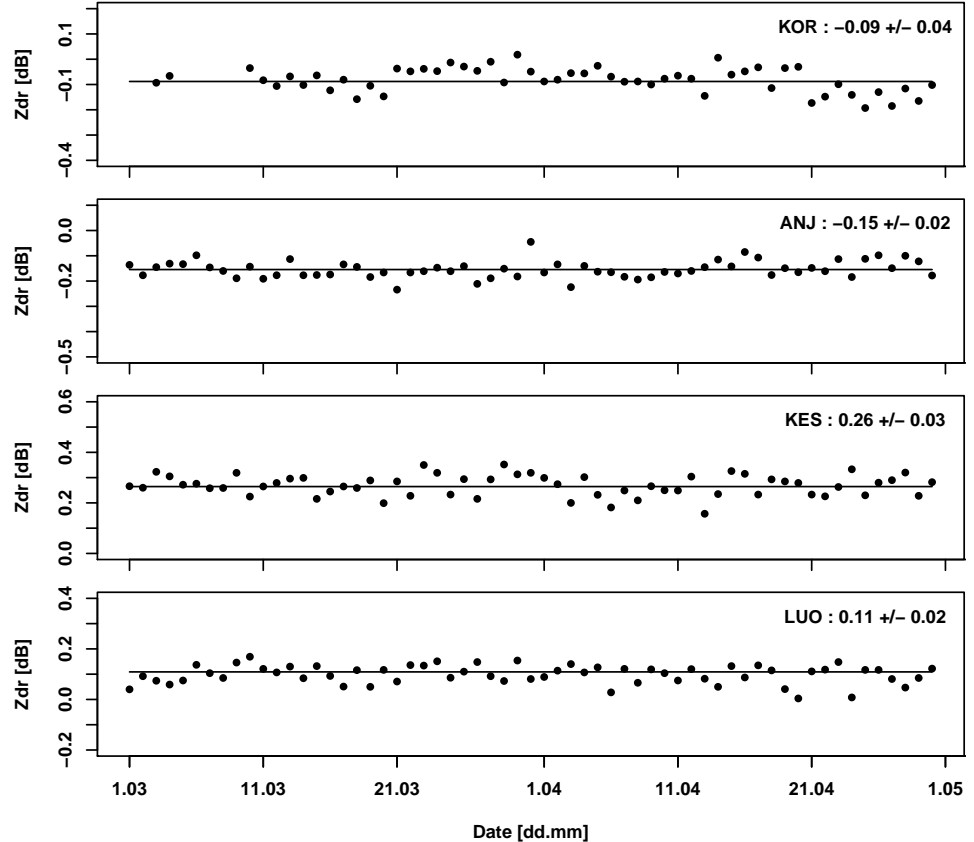

**Figure 8.** $Z_{dr}$ results for March and April of 2015 for FMI radars indicated in the panels. The numbers in each panel give the mean and standard deviation for the April data.

The additional benefit of doing the fitting to both $Z_H$ and $Z_V$ is that one can determine the squint angle, i.e. the angle between the pointing directions of the two polarizations. The two upper panels of Fig. 7 show statistics of the elevation and azimuth pointing results of the H and V polarizations for one month of data for the Anjalankoski (ANJ) radar. The analysis confirms that the H and V beams are well aligned. The squint angle for ANJ is less than 0.02°. A similar analysis of all the radars confirms that for most other radars in the network the squint angle is less than 0.01°. These squint angle values are similar to those reported earlier by Frech (2013) for the German network.

The lower panels compare the width values of the two polarizations. There is a clear difference in the image widths so that for the V polarization the image is wider in elevation and for the H polarization in the azimuth. This is a results of the antenna design, and the results are typical for the whole network, and are consistent with the width values given in Table 1.





Figure 8 shows $Z_{dr}$ results from four polarimetric FMI radars. The $Z_{dr}$ values are differences of the horizontal and vertical powers, which were obtained by doing a 5-par fit to both polarizations separately, as recommended above. Figure 8 shows that the standard deviation during the last 30 days is 0.02° for two radars and 0.04° or less for all radars. This is an indication of the stability of the radar system itself and also of the analysis method. The standard deviations are significantly lower than 0.2 dB reported by Holleman et al. (2010a), or 0.05 dB reported by Gabella et al. (2015), probably because due to the better rain rejection in the estimation of the hits. The $Z_{dr}$ bias itself varies from radar to radar.

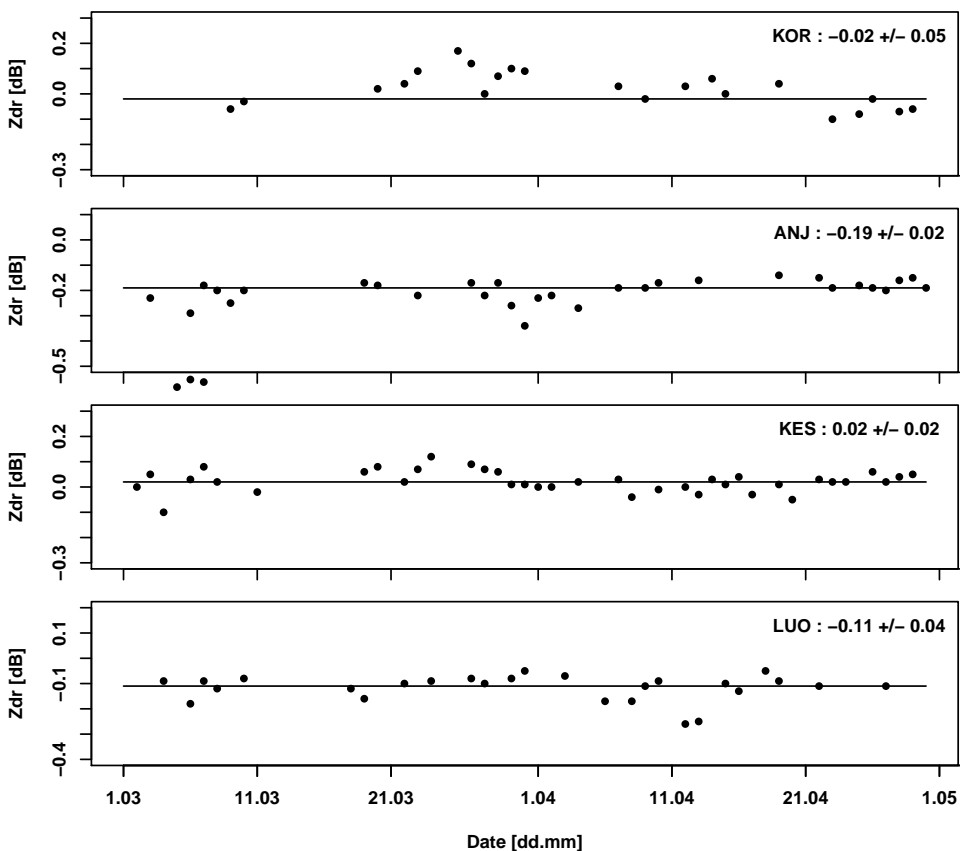

**Figure 9.** $Z_{dr}$ results in rainfall from the zenith scan for March and April of 2015 for FMI radars indicated in the panels. The numbers in each panel give the mean and standard deviation for the April data.



### 4.3 $Z_{dr}$ results from the zenith scans in rainfall

Figure 9 shows $Z_{dr}$ bias values based on the zenith scan measurements for March and April 2015 for the same radars as used in Fig. 8. The results are not available for all days, because the determination requires rain above the radar. Yet the bias has been determined for more than half of the days. The figure shows that the $Z_{dr}$ bias varies from radar to radar but that it is remarkably

stable as indicated by the standard deviation of the data. These bias values are not directly comparable to those from the sun for various reasons. The zenith scans are used to determine a system $Z_{dr}$ bias which is entered into the signal processing and subtracted from all $Z_{dr}$ measurements, and hence also from the data containing the solar hits. An effect of the change in the system bias is seen in Fig. 6 in which the $Z_{dr}$ values make a downward step on August 21. Also the $Z_{dr}$ values based on solar hits include the receiver chain only, and hence the difference on the transmitter losses and antenna gain between the horizontal

and the vertical channels, given in Table 1 need to be added before the comparison. When the above is taken into account, the difference in absolute value between the two methods is about 0.1 dB for four radars in the network, around 0.4 dB for three radars, and larger for one. Noting the stability of the results from both the solar and zenith methods, and assuming that the antenna gains do not change over time, the most obvious explanation is that the transmitter losses are not correct.

The fact that the transmitter loss values might be incorrect does not affect the usability of the $Z_{dr}$ measurements, because

the zenith scan measurement takes all loss factors into account, and the bias determined from there corrects for all possible errors in the loss or gain difference values. However, any error in the losses biases the reflectivity values, and also limits the use of the solar based $Z_{dr}$ values to that of monitoring of the stability of the receiver chain. In the latter they are most valuable, because an estimate is obtained during most of the days. The solar $Z_{dr}$ values also provide a consistency check, assuming that the $Z_{dr}$ bias correction is done correctly. If the solar $Z_{dr}$ values are then not close to zero, it is highly probable that the losses,

either receiver or transmitter or both, are not correct or there is an error in the antenna gain figures. On the other hand a value close to zero is no guarantee that all the above are correct, because several errors may cancel each other.

### 5   Conclusions

The differential reflectivity of the quiet sun is zero and constant over the solar surface but the radar measurements include also the effect of the antenna and the receiver chain. In case the beams shapes of the horizontal and vertical polarizations were fully

identical, all $Z_{dr}$ observations would have a constant value over the solar surface. This is not the case as shown by examples in Holleman et al. (2010a) and in the present paper. Hence the $Z_{dr}$ bias cannot be estimated accurately by taking a simple median or mean over all solar $Z_{dr}$ observations. Instead it is necessary to fit the observations to a model. The most stable and recommended method is to make a full 5-parameter fit to both polarization channels separately, which works well also when the surface approaches constant which corresponds to infinite width values. In that case direct fitting to $Z_{dr}$ would come

increasingly difficult. In case the widths of the two polarizations differ, it is possible to perform a 5-par fit to $Z_{dr}$ or to use the antenna pointing determined from the reflectivity fitting and fit for the remaining three parameters.

The zenith measurements of $Z_{dr}$ in rain include both the transmitter and the receiver chain, whereas the solar measurements include only the receiver. The zenith measurements are essential because they are used to estimate the $Z_{dr}$ bias, which is





subtracted in the signal processing from all $Z_{dr}$ measurements. The solar $Z_{dr}$ observations are valuable for the monitoring of the receiver stability, and they also provide a consistency check on the measurements of the transmitter losses and gains. If the solar $Z_{dr}$ bias is not close to zero, when the zenith bias correction is applied, the transmitter losses and gains are suspect.

The statistical accuracy of the $Z_{dr}$ results, both solar and zenith observations, is better than 0.02 dB for most radars, based on
the analysis of on month of data. This is a significantly better value than those reported earlier (Holleman et al., 2010a; Frech, 2013; Gabella et al., 2015). The accuracy is a combination of the radar system performance and that of the analysis system. In the latter we have used a number of existing methods (Huuskonen and Holleman, 2007; Holleman et al., 2010a) and introduced a number of new tools. With this we have developed a method to extract solar hits accurately without any significant rain and clutter contamination. Noting that the $Z_{dr}$ from the sun and the signals we want to remove, e.g. clutter and rain, may be close
to each other in value, we have introduced a method where the quality control is done using the reflectivity data. The improved level of accuracy provided by these methods allows us to monitor and detect changes in the receiver chain much better than before. All this improves the quality of $Z_{dr}$ data, which is most important for many polarimetric algorithms. It is recommended that this improved online monitoring of the differential reflectivity is performed daily for all polarimetric radars in the network, preferably in combination with the rain calibration at zenith.

*Acknowledgements.* The numerical analysis and the figures have been prepared using the R software (R Core Team, 2012).



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
