# Peer review of "IMPROVED ANALYSIS OF SOLAR SIGNALS FOR DIFFERENTIAL REFLECTIVITY MONITORING"

_Atmospheric Measurement Techniques, 2016_

## Referee Comment (RC1) · Anonymous Referee #1 · 11 Mar 2016

Review of "Improved analysis of solar signals..." by A. Huuskonen et al. 2016.

The authors describe procedures for obtaining the system ZDR bias and quint angle using solar radiation. The paper contains valuable data and its results can be used for the monitoring of ZDR calibration in operational weather radars. To make the authors' findings clearer, I'd recommend considering the following issues.

1. It was not clear to me how ZDR of the solar flux has been obtained in the Finnish radars. In the calculations of ZDR, the IRIS subtracts the system ZDR from ZDR measured in all range gates (see also page 15 lines 6, 7) including the gates with solar hits. Let denote the ZDR biases introduced by the three major radar components, i.e., the transmitter, receiver and antenna, as $B_T$, $B_R$, and $B_A$ correspondingly. The authors consider the antenna bias as a component of the receive bias, i.e., $\Delta R_{dr} = B_R + B_A$ in eq. (17). It is not clear from the manuscript whether the authors consider the antenna bias as a part of transmitter bias as well, i.e., $\Delta T_{dr} = B_T + B_A$. I assume this here. The system ZDR ($ZDR_{sys}$) is

$$ZDR_{sys} = \Delta R_{dr} + \Delta T_{dr} = B_R + B_A + B_T + B_A = B_R + 2B_A + B_T. \qquad (A)$$

This is eq. (17) rewritten in more detail. Now consider the measurements in the solar flux. The radar processor subtracts the system ZDR from measured ZDR values. ZDR from the sun is $B_R + B_A$ before subtracting $ZDR_{sys}$. So the reported ZDR from the sun ($ZDR_{sun}$) is:

$$ZDR_{sun} = B_R + B_A - ZDR_{sys} = -B_A - B_T = -\Delta T_{dr} \qquad (B)$$

Eq. (B) shows that the reported ZDR depends on the bias in transmit. $ZDR_{sun}$ depends on the bias in receive as well. Thus the reported solar ZDR depends not only on the bias in receive as the authors stated throughout the text but on the transmit bias as well. If $B_R$ changes and $ZDR_{sys}$ have not been yet adjusted by rain measurements, then $ZDR_{sun}$ changes as well. So the reported $ZDR_{sun}$ depends on the receiver and transmitter biases. Please clarify if this is correct.

2. The ZDR scatterplot from the solar spikes shown in Fig 2 (the right panel) exhibits quite strong diagonal disturbances. Such a feature has not been observed in French, German, and US radars. To make this feature more pronounced, a scatterplot from a distinct solar scan is desirable. Could you please show data from a box solar scan when the antenna scans the solar disk. Such data can be obtained with the IRIS routines by setting up a sector scan with an angular step of 0.2 deg . Such data could already exist in the radar data archive.

A "saddle" ZDR scatterplot makes it difficult to match it with a parabolic surface eq. (8). The ZDR surface is a difference of two parabolic surfaces eq.(2), i.e., it should be a parabolic surface as well. The observed ZDR saddle surface raises a question of its origin. I wonder if this is a feature of the antenna. Please compare the ZDR diagonal disturbances with the placement of the antenna struts that support the feedhorn. Are there 4 antenna struts placed about 45 deg to the horizon?

3. It is recommended in the manuscript to obtain the system ZDR by subtracting the fitted powers from the sun in H and V channels. The radar reports $Z_H$ and ZDR from which $Z_V$ is obtained as $Z_V = Z_H$ – ZDR (eq. (12) and also p.7 line 8). It is stated (p.7 line 8) that this implies $C_H = C_V$. The latter means that the system ZDR is zero dB, which can be not the case for radar. A nonzero system ZDR implies that $C_H$ differs from $C_V$ because amplifications and losses in the polarization channels are different. So it is not clear from the manuscript how the system ZDR bias affects the calculation of $Z_V$.

   It is being recommended obtaining the ZDR bias by subtracting the fitted solar powers in the channels (page 12). Then what is the purpose of section 3.3 where the modeled ZDR signature is considered?

4. Measurements in rain with a vertically pointed antenna could be affected by a water film on the radome. A rain film on a radome is not ideally uniform that leads to different attenuation of H and V waves. For low antenna elevations, ZDR can be distorted by more than 1 dB at a wet radome (e.g., https://ams.confex.com/ams/96Annual/webprogram/Paper288057.html ) Similar effect could be expected at vertical incidence. So I wonder if ZDR from rain at vertical incidence is so perfect.

Some other comments.

It follows from Fig. 3 that rain has been observed at altitudes as high as 8 km. These are very high altitudes for Finland. The radar volumes at such heights are above the melting layer, most likely. Radar UTA in the figure is absent in Table 1.

After transmitting a radar pulse, radar receivers can be out of their normal stage during a time interval equivalent a range of 8 km (page 8 line 10). Most likely, no rain can be present at this height. How do you calibrate ZDR in such situations?

Signals with SNR > 5 dB have been used in the analysis (page 8 line 13). Has noise been subtracted from the measured powers? What is SNR of the sun flux and how noise is processed in the solar hits?

---

## Referee Comment (RC2) · Anonymous Referee #2 · 16 Mar 2016

**Review of manuscript: amt-2016-52**

**Title: Improved Analysis Of Solar Signals for Differential Reflectivity Monitoring**
**Authors: Asko Huuskonen, Mikko Kurri and Iwan Holleman**

**GENERAL COMMENTS**

This manuscript reviews and improves several aspects of the methodology that uses solar hits found in operational scans for daily monitoring of the calibration of the weather radar differential reflectivity (ZDR) bias in reception. The main achievements are the proposal of a procedure to improve the accuracy of the power and differential reflectivity estimated for the solar hits and that the authors provide solid arguments supporting a separate fit to solar hits' power in each channel, in contrast to a direct fit to solar ZDR data. The authors also give good guidelines about how to interpret the solar ZDR bias results in comparison to zenith ZDR measurements in rain in order to identify potential inconsistencies in the transmitter or gain losses. I believe that this work is of interest for the weather radar community and fits well within the scope of AMT journal. Comments, questions and suggestions to be considered by the authors are listed in the following.

**SPECIFIC COMMENTS**

- Eq. (1) given in P.3 is valid when the calibration reflectivities $dBZ_0^{H,V}$ are available for both channels. In this case both radar constants $C_H$ and $C_V$ are known - note that the radar constants $C_H$ and $C_V$ include antenna parameters and transmission losses regarding their respective polarisations.
  However, often the radar processor only allows for a single calibration reflectivity and the differential reflectivity offset, $dBZ_0^H$ and $ZDR_0$ (e.g. $Z_{dr}^{rain}$), for instance. In this second case, a single radar constant is used for the two channels (e.g. $C_H$) and the differences between radar constants are accounted for in the $ZDR_0$ calibration. If the measurands available are $Z_H$ and $Z_{dr}$ and only $C_H$ is known (the difference $C_H - C_V$ is accounted in the calibration $ZDR_0$), $P_H$ can be calculated from Eq. (1) but it is not possible to calculate $P_V$. Instead the following quantity can be calculated:

  $$P_V - (C_H - C_V) = P_H - Z_{dr} = Z_H - Z_{dr} - C_H - f(r)$$

  Therefore, Eq.(12) also holds valid for this case and it is not correct to say that $C_H = C_V$ (P.7, L.8). In this case, separate fits would mean fitting $P_H$ and $P_V - (C_H - C_V)$ but the difference between the peak powers would still give an estimate of the differential receiver bias.

  I think that these two cases need to be more explained and the procedure to follow in the two fitting approaches (direct $Z_{dr}$ fit and separate H/V fit) explicitly described for each case.

- If I am not mistaken, the $\hat{Z}_{dr}$ value corresponding to perfect antenna-sun alignment estimated from the fit has a slightly different interpretation depending on wether $Z_H$ and $Z_V$ are separately available with their own calibration reflectivities (Frech, 2013) or wether $Z_H$ and $Z_{dr}$ are available (Holleman et al., 2010a) together with the calibration $dBZ_0^H$ and $ZDR_0$. In the first case, $\hat{Z}_{dr}$ carries information about the differential sensitivity (i.e. differences in noise figure between channels) while in the second case $\hat{Z}_{dr}$ gives information (except for the $ZDR_0$ calibration value) about the bias of the linear depolarisation ratio or differential receiver bias. I have tried to prove this in the Appendix at the end of this document.

- **P.7, Eqs.(15)-(16):** Since the squint angle is studied in an upcoming section, it is important to mention that these equations are derived assuming that the pointing biases are equal for both polarisations. Also Eqs.(10)-(11); the parameters $B_{\phi,\theta}$ give the position of the minimum/maximum/saddle point of the surface but represent the biases only under the aforementioned assumption.

- **P.6-7, Section 3.3:** I suggest that this section is reordered for clarity and rigour; it should start with Eq.(12) to arrive at Eq.(8), since the latter model is the result of subtracting the two paraboloid surfaces. Then, Equation sets (9)-(11) and (13)-(16) could be joined in a single set. Having Eqs.(13)-(14) before the sentence in lines 18-19 would also be helpful. In addition, the first paragraph of the section may be moved after the mathematical derivation so that is easier to understand the qualitative description of the left panel in Figure 2.

- **P.8, Fig.4:** The median estimate (red) and the mean estimate (green) lines overlap but their corresponding window upper and lower widths are not at the same distance from each other. There seems to be an inconsistence in the green lines drawn for the mean.

- **P.9, L.8-9:** I could not find the solar hit rejection explained in sentence "In addition solar hits with standard deviation ..." in the references given at the beginning of the paragraph. Do you apply it for the present work only? If so, maybe it should be explained or this sentence should be placed later in the section. Also, what is approximately the standard deviation expected for a solar hit?

- **P.9, L.11-12:** Just a "picky" comment: the method of removing outliers using the fit curve as reference may fail also when there is a single outlier but "badly located" so that it alone biases the fit curve. This is implied in the sentence just afterwards "The results are further improved ..." but may be interesting to mention it explicitly.

- **P.10, L.26:** For clarity, this sentence may be extended to explain that, after the estimation of the width and the filtering, the mean value for ranges > 50 km is computed.

- **P.10, L.28-30:** Another reason to use a fixed width may be that at low elevations there might be too few (or none, depending on the maximum range) range bins at high altitudes available for estimation of the filtering window width.

- **P.10, L.30:** Could you give a value for the fixed window width recommended for filtering (e.g. the one estimated using data at high altitudes in the analysis in Fig. 4)?

- **P.9-10, Section 4.1:** I think it would be interesting to provide further evidence on how the proposed filtering increases the number and quality of the solar hits, to support what is stated in P.2 L.15-16. For example, are the fit estimates more stable or the RMSE of the fit lower when applying the proposed quality control? This is somewhat accomplished in P.14, L.2-6 but comparisons of the number of hits and standard deviation for the same dataset before and after the quality control may be desirable.

- **P.11, L.7-9:** I think the 3-parameter fit with fixed pointing may give biased results if there is a significant squint angle (from the reasoning in the comment above for P.7, Eqs.(15)-(16)).

- **P.11, L.12:** In this sentence it is not clear which methods are compared. The first method described (the one used in Holleman et al. (2010a)) is not analysed. Maybe this should be explicitly mentioned in case the reader expects to find it among the methods compared in Fig. 6.

- **P.12, L.5-6:** The sentence "For surfaces ... is one of the fit parameters" is a very good point and hints to a very strong argument to support the separate fit for the horizontal and vertical channels. Even if it is further discussed in the conclusions, I think it would be beneficial to put more stress in this reasoning at this point of the manuscript. The case described (constant surface) corresponds to an extreme case of ill-conditioning of the inverse problem in Eq.(8) (there are more parameters in the model than needed to explain the observations). This example also indicates that fitting Eq.(8) is not an appropriate methodology for routine usage, since the well-conditioning of the problem depends on the magnitude of the differences between the horizontal and vertical widths, which may change in time and from radar to radar.

- **P.11-14, Section 4.2:** Could you specify which measurands were available for the calculations in this section?

- **P.15, L.16-17:** If I am getting it right, assumed that the antenna gains are correct and that the bias corresponds to inaccurate transmitter losses, then this bias is systematic for the reflectivity values and should not affect the estimation of the pointing errors or the squint angle.

**TECHNICAL CORRECTIONS**

**P.1, L.1:** Suggest something like "refined" or "optimised" instead of "developed"

**P.1, L.3:** Suggested, for clarity, something like: "... rain and clutter contaminated gates ... "

**P.1, L.4-5:** I think that it is not clear to which analyses this sentence refers

**P.1, L.6:** "differential reflectivity offset/bias"?

**P.1, L.13:** Suggested: "Several methods for $Z_{dr}$ calibration exist ..."

**P.1, L.23:** "normal radar operations do not need to be stopped"

**P.2, L.1:** Suggested: "antenna alignment information"

**P.2, L.3:** "introduced the on-line method for the solar"

**P.2, L.6:** Suggest to remove the comma after "pointing"

**P.2, L.22:** "where the six elevations" ? Or are there more elevations below 9° that are not scanned at single PRF?

**P.2, L.22:** "Every 5 minutes"

**P.2, L.25:** Suggest to add a comma after "For convenience"

**P.2, L.24-25:** Suggest to change the order of the sentence for clarity: "now two new radars have been added to the network and all radars except one are polarimetric"

**P.2, L.29:** Suggest to add a comma after "method"

**P.3, L.1:** Suggested: "increase with range"

**P.4, L.5:** "which is slightly"

**P.4, L.5:** Maybe a brief explanation here of why the distribution is wider in azimuth?

**P.5, L.1-3:** Suggest moving these sentences "We assume ... by the least squares method" to L.6 after "... vertical polarisations" and before "The elevation width ...", because the relation of the parameters to the widths and biases is explained afterwards.

**P.5, L.5:** Suggested: "... these parameters may be different ..."

**P.6, L.3:** Suggested: "the distribution has the form"

**P.6, L.8:** Lacks a space after the point

**P.6, L.20:** The "powers" term in this sentence may lead to confusion since Eq.(2) is a polynomial equation. Maybe something like "we can substitute the powers in Eq.(12) using Eq. (2)"

**P.7-9:** Suggest to structure subsection 3.4 as a separate section; e.g. one section dealing with sun calibration and the next with rain calibration.

**P.8, L.10:** "Because of the transient effect"

**P.10, L.4:** Lacks a space after the point

**P.10, L.7:** "devise" here instead of "device"

**P.10, L.9:** Suggested: "and this power together with ..."

**P.10, L.17:** Do you mean "rain contamination" here?

**P.11, L.5:** Is it Eq.(8) here?

**P.14, L.3:** The degree units should be dB instead

**P.14, L.4:** "than the 0.2 dB"

**P.14, L.5:** "or the 0.05 dB"

**P.14, L.5:** "probably due to ..."

**Appendix**

Solar powers at the receivers:

$$t_H = g_H^r \, p_H + N_H$$
$$t_V = g_V^r \, p_V + N_V \tag{1}$$

Calibration reflectivities:

$$dBZ_0^H = 10 \log \left( \frac{c_H \, N_H}{g_H^r} \right) = 10 \log \left( c_H \, I_0^H \right)$$
$$dBZ_0^V = 10 \log \left( \frac{c_V \, N_V}{g_V^r} \right) = 10 \log \left( c_V \, I_0^V \right) \tag{2}$$

And differential reflectivity offset:

$$ZDR_0 = 10 \log \left( \frac{g_H^r \, c_V^t}{g_V^r \, c_H^t} \right) \tag{3}$$

The radar constants include the transmission gains. The following results are derived for the $Z_{dr}$ fit but they also apply for separate H/V fits in both cases.

**case 1: Direct $Z_{dr}$ ($ZDR_0$)**

$$Z_{dr} = 10 \log \left( \frac{t_H - N_H}{t_V - N_V} \right) - ZDR_0$$
$$= 10 \log \left( \frac{p_H}{p_V} \right) + 10 \log \left( \frac{g_H^r}{g_V^r} \right) - ZDR_0 \tag{4}$$

Therefore, if $ZDR_0$ is added to the $\hat{Z}_{dr}$ fit estimate, an estimate of the linear depolarisation ratio offset ($XDR$) is obtained.

**case 2: $Z_H$ ($dBZ_0^H$) and $Z_V$ ($dBZ_0^V$)**

$$Z_H = 10 \log \left( \frac{t_H - N_H}{N_H} \right) + dBZ_0^H + f(r) \tag{5}$$

$$Z_V = 10 \log \left( \frac{t_V - N_V}{N_V} \right) + dBZ_0^V + f(r) \tag{6}$$

$$Z_{dr} = Z_H - Z_V = 10 \log \left( \frac{p_H}{p_V} \right) + 10 \log \left( \frac{N_V}{g_V^r} \frac{g_H^r}{N_H} \right) + dBZ_0^H - dBZ_0^V \tag{7}$$

$$= 10 \log \left( \frac{p_H}{p_V} \right) + 10 \log \left( \frac{I_0^V}{I_0^H} \right) + dBZ_0^H - dBZ_0^V \tag{8}$$

In this case, if calibration reflectivities and antenna gain ratio are subtracted from the $\hat{Z}_{dr}$ fit estimate, the estimate obtained is the ratio of the equivalent front-end noises $\frac{I_0^V}{I_0^H}$.

---

## Referee Comment (RC3) · Anonymous Referee #3 · 23 Mar 2016

The authors should make clear the limitations of the bias calculations from the sun scan measurements. From the sun one can establish the overall gain bias, but the antenna effects although included are not equivalent to the antenna effects that are present in the two way measurements. There are some key assumptions that underpin bias measurements with the sun scan. These are not articulated in the paper. Here I list the constraints.

The measurement of sun flux for calibrating Zdr implies that in the ratios of powers at the two polarizations due sole to the antenna effects

$$\frac{P_h(antenna\ one\ way)}{P_v(antenna\ one\ way)} = \frac{g_h \int_\Omega s_h(\theta,\phi) f_h^2(\theta,\phi) d\Omega}{g_v \int_\Omega s_v(\theta,\phi) f_v^2(\theta,\phi) d\Omega} , \tag{1}$$

can be estimated precisely. $\Omega$ is the solid angle centered on the sum (on which the beam is centered too), integration is mainly over the main beam, the $s_h(\theta,\phi)$ is the distribution of sun flux within the beam for the H polarization, $s_v(\theta,\phi)$ is for the V polarization; on the average these are equal; $g_h$ and $g_v$ are gains, and $f_h^2(\theta,\phi)$, $f_v^2(\theta,\phi)$ are one way power pattern functions. Equation 1 is part of the computation of the differential reflectivity (hence bias) in the proposed method. The actual antenna bias is due to the two way antenna effect. The pertinent equation is

$$\frac{P_h(antenna\ two\ way)}{P_v(antenna\ two\ way)} = \frac{g_h^2 \int_\Omega f_h^4(\theta,\phi) d\Omega}{g_v^2 \int_\Omega f_v^4(\theta,\phi) d\Omega} \tag{2}$$

The bias on receive $Z_{dr}^{(R)}$ can be conveniently written as

$$Z_{dr}^{(R)} = C_r \frac{g_h \int_\Omega s(\theta,\phi) f_h^2(\theta,\phi) d\Omega}{g_v \int_\Omega s(\theta,\phi) f_v^2(\theta,\phi) d\Omega} = C_r \frac{g_h K_{h1} B_{h1}}{g_v K_{v1} B_{v1}} \tag{3}$$

The $C_r$ is receiver bias and $B_{h1}$, $B_{v1}$ are beam cross sections, say at the 6 dB levels below the peak. Note that the assumption behind (3) is that the main lobe of the beam is well represented with a two dimensional Gaussian function such that the beam cross sections at the horizontal and vertical polarizations are related to the integrals via

$$K_{h1} B_{h1} = \int_\Omega s(\theta,\phi) f_h^2(\theta,\phi) d\Omega, \ \text{and} \ K_{h2} B_{v1} = \int_\Omega s(\theta,\phi) f_v^2(\theta,\phi) d\Omega ; \tag{4}$$

$K_{h1}$, $K_{h2}$ are constants of proportionality that relate the integrals to the beam cross sections, and the beam is centered on the sun.
From the radar equation the measured $Z_{dr}$ assuming homogenous isotropic scatterers is

$$Z_{dr} = C_t \frac{g_h^2 \int_\Omega f_h^4(\theta,\phi) d\Omega}{g_v^2 \int_\Omega f_v^4(\theta,\phi) d\Omega} C_r = C_t \frac{g_h^2 K_{h2} B_{h2}}{g_v^2 K_{v2} B_{v2}} C_r = C_t \frac{g_h^2 K_{h2} a_h B_{h1}}{g_v^2 K_{v2} a_v B_{v1}} C_r . \tag{5}$$

Here $C_t$ is the bias on transmission and it excludes the antenna effect. It is assumed that the beam is filled with uniformly distributed scatterers. $B_{h2}$, $B_{v2}$ are the cross sections of the two way antenna patterns and $K_{h2}$, $K_{v2}$, are the proportionality constants relating the beam cross sections to the integrals of the two way patterns. For Gaussian patterns it follows that $B_{h2} = a_h B_{h2}$, where $a_h$ is a constant; similarly $B_{v2} = a_v B_{v2}$. For convenience write (5) as

$$Z_{dr} = C_t \frac{g_h K_{h2} a_h}{g_v K_{v2} a_v} \frac{g_h B_{h1}}{g_v B_{v1}} C_r . \quad (6)$$

The bias due to the receiving path $\dfrac{g_h B_{h1}}{g_v B_{v1}} C_r$ lacks the proportionality constants $K_{h1}$, $K_{v1}$ which are in (3) and are estimated from the sun scan. But these two should be very close. On the transmission side the antenna part $\dfrac{g_h K_{h2} a_h}{g_v K_{v2} a_v}$ is not known. This ratio should be very close to 1. The constants of proportionality for the H and V parts should be equal. And if the gains $g_h$, $g_v$ are close, the sun calibration can indeed provide a good estimate of the bias. Note that the beam cross sections used in the assumptions need to be equal but need not be completely overlapping. For example two elliptical cross sections could have major axis not aligned, for example orthogonal. One can quantify (or set limits) on how well can the sun scan be used for calibration by comparing the $g_h B_{h1}$ to $g_v B_{v1}$ (take the ratio in dB). That is done next for the five radars.

Going back to the paper I examined the values from table 1. So I compute the $G_h - G_v +$ 10*log[H(el,H) H(az,E)]-10*log[ V(el,E) V(az,H)]; that is $(g_h B_{h1}/ g_v B_{v1})$ in dB scale and obtained: -0.045, -0.03, -0.182, 0.12, -0.015 for the five radars respectively. Thus I conclude that for the radars No 1, 2, and 5 the sun scan could provide bias to within ± 0.1 dB.

Some other comments:

If the radar is scanning the sun the position is wider in azimuth because the effective beamwidth is larger.

You state "uncorrected". It is clearer to indicate that "Ground clutter had not been applied to the data, and noise power has not (or has?) been removed." Uncorrected is confusing and strictly would mean that you had correct data and then you "uncorrected it".

Page 14, line 5: Remove "because"
Page 16, line 5: "one months of data"

Fig. 5 not clear – Caption states "Probability" whereas on the axis I see "Power fraction".

Caption Fig. 2 end sentence you repeat the.

Sentence on page 5 "In addition solar hits with standard deviation much larger than that expected for a solar hit (e.g. three times larger) have been rejected." Is this really what you are doing? If

so how are you computing the standard deviation, local running average in azimuth at a constant altitude, or else?  Or, do you really mean the data that deviate by thee standard deviations from the mean are discarded?

---

## Referee Comment (RC4) · M. Gabella (Referee) · 31 Mar 2016

**A review of "IMPROVED ANALYSIS OF SOLAR SIGNALS FOR DIFFERENTIAL REFLECTIVITY MONITORING" by Asko H. et al., submitted to AMT (amt-2016-52)**

The paper describes a number of analysis methods for monitoring the residual differential bias between horizontal and vertical reflectivities of solar signals. Personally, I find it interesting and stimulating. I also believe that is of interest not only for the academic world, but also for the operational meteorological radar community, since dual-polarization receivers are now available in several Countries. The paper shows nice, intuitive and informative picture such as Fig. 2, 3, 6, 8 and 9. Maybe it could be made a bit shorter and more concise? I am confident that by doing so, it will attract more readers. Aiming at this, I would suggest, for instance, to shorten in particular Sec. 4.1, by deleting also Figures 4 and 5. Finally, it is probably worthwhile warning the readers about the (relatively weak) intensity of the Solar(+Noise) Signal: ~10 dB SNR with high-gain and high-sensitivity radar as long as the beam axis is hitting the center of the solar disk; but this is not the case for most of the hits acquired during operational monitoring …

Several other specific comments and suggestions to be considered by the authors are listed below.

Locarno Monti, 31.3.2016

Yours faithfully,
Marco Gabella

**Fundamental, interesting, "hot" topics [from my (biased!) viewpoint]**

Page 3 Eq. (1), term $-2ar$ , which is two-way gaseous attenuation: is the term $a$ in your Radar Signal Processor set to a constant values, independently of altitude? Curiosity: what value of $a$ (in dB/km) at the sea level do you use?

Page 7, just after Eq. (16); I would re-write the sentence in a completely different way; I mean, your great idea of using solar hits during operational scan for monitoring weather radar performances, has been indeed useful for many national weather services … Hence, I would go along that tradition by changing the sentence into a recommendation. For instance: unfortunately some radar software systems do not provide both H and V reflectivities, but instead only the H reflectivity and the differential one. In this way, when deriving $Z_V$ from the two previously-mentioned quantity, one has to assume that $C_H = C_V$, which is not necessarily always the case. Or something similar; with this kind of formulation, future dual-polarimetric radar users will always deal with $Z_H$, $Z_V$ and $Z_{DR}$ and not only with two of them; there are other advantages associated to this choice: for instance, $Z_{DR}$ can be set to 0 dB by introducing an appropriate offset, while leaving $Z_H$ and $Z_V$ untouched and calibrated at their best according to the Probert Jones Equation (nominal values of Tx, Rx Losses and antenna Gain).

Page 8: paradoxically, you just briefly mention the important issue of Signal-to-Noise ratio in Sec. 3.4 ("Calibration of $Z_{dr}$ during rain"), which deals with "RELATIVELY STRONG" hydrometeor signals and you do not mention it when dealing with "much WEAKER" solar signals. On the one hand, please, consider a 10 dBZ echo say at 10 km altitude (when vertically pointing): SNR is of the order of ~20 dB for the Finnish high sensitivity radars. On the other hand, even when the antenna beam axis is hitting the center of the solar disk, the Sun+Noise "signal" is just ~10 dB stronger than the Noise; even worse, during the operational scan, the beam axis rarely hits the center of the Sun, so that typical SNR is probably around ~8 dB or less … I ask to authors to mention this important aspect and warn the readers about it: maybe you could anticipate it in the introductory part? Another important point: here at MeteoSwiss, ZH and ZV values are derived after Noise subtraction, so that solar flux estimates "back-retrieved" from reflectivity values in the PPI are "intrinsically noise subtracted". Probably, the same is at FMI and it is worthwhile to inform the readers about it.

Eq. (17) and Fig. 6 (linked with Fig. 8 and page 15, see my next comment): I would plot data only from Aug. 1 to Aug 20, before the discontinuity that is only subsequently described at page 15, line7-8. Alternatively, I would anticipate here such sentence, maybe even in the caption. More important: am I missing something? By simply looking at the picture, I would conclude that Zh(5par fit)-Zv(5par fit) [CYAN curve] is BIASED of approximately +0.1 dB with respect to the median (GREEN curve), which in turns look unbiased. Why? Curiosity: what is the st. dev. of these first 20 points? (it looks of the order of 0.05 dB or less…) Adding up, sorry to say, from Fig. 6, I do not conclude that 5-param-fit is better than median. Why do you conclude that the median is not recommendable? Finally, do you have explanation for the mean having a negative BIAS with respect to the median?

Page 14 (text related to Fig. 8); line 3: dB instead of degree; line 5: yes, MeteoSwiss figures (0.05 dB for Albis, 0.06 dB for Lema and Dole, Gabella et al. 2015) were obtained using both wet and dry days; however, as stated at page 51, Sec. 2, of Gabella et al. 2015, "In the MeteoSwiss approach, only the last 5 km (60 gates) of each radar bin are averaged for Sun power retrieval"; this means that our solar echoes are at stratospheric and mesospheric altitude … Probably here it is worthwhile to emphasize that MeteoSwiss figures have been obtained: A) by neglecting atmospheric attenuation (affecting in a different way the 15 different elevations used); B) by simply using the median, instead of Zh(5par fit)-Zv(5par fit).

Fig. 8: could you please add μ±σ also for March data? (e.g. in the upper left part of each picture) Could you please add a picture for UTA radar?

Fig. 9: could you please add μ±σ also for March data? (e.g. in the upper left part of each picture) Could you please add a picture for UTA radar?

Page 15, line 12-13. This is indeed an important and interesting statement and a key point of the paper, I think; you could be less concise and try an interpretation of your results. In terms of reproducibility and repeatability, I am confident that μ±σ for March data will help the interpretation. (You could even add more months…) However, in order to make conclusions regarding this interesting point you should also provide a table with H and V Tx Losses and antenna Gain.

**Conclusion**, page 16, line 4-6. Here, I would list fact and figures in a more neutral and specific way: instead of "statistical accuracy … is better than 0.02 for most radars" I would write "the standard deviation of differential reflectivity for 30 daily samples (April 2015) is 0.02 dB for two radars, 0.04 and 0.05 dB for the other two radars" or something similar … [will you add also the 5th UTA radar, please?]. "This is significantly better of what reported earlier, namely 0.2 dB for the MeteoFrance Trappes radar [and then please specify 90 days; by the way, was it Zh(5par fit)-Zv(5par fit)? Maybe you can split the Trappes 90 days in three monthly subsets and tell us it st. Dev. is ~0.2 dB for all three subsets or whether it varies (I hope and I am confident it does not… )

Similarly, you can explicitly mention the DWD figure (Frech, 2013): what are the number of days and constraint used? … (5-par? Or median? atmospheric attenuation correction included?) and MeteoSwiss ones. As stated at the top of the page (no atmospheric correction, simply using the median, not the 5-par fit … ) The number of days used for the 3 MeteoSwiss radar was larger than two hundreds (see Table 4, Gabella et al. 2015).

Finally, I think it is worth mentioning other examples; for instance, as reported in Gabella, M.; Huuskonen, A.; Sartori, M.; Holleman, I.; Boscacci, M. The solar slowly varying component as detected by dual-and single-polarization C-band weather radars in Europe, Electromagnetic Wave Propagation and Scattering in microwave Remote Sensing, Technische Universität Chemnitz Publikationen, submitted on September 29, 2015

during a common observation period of 100 days in 2014, the dispersion of the difference between H and V, namely the standard deviation of 100 values of 10 Log(H/V), was 0.06 dB for Albis and 0.08 dB for ANJ (see Table 3 in the above-mentioned paper). While using the median instead of 5-par fit for

**ANJ**, it was **0.12 dB** ; also this other example shows that using 5-par-fit gives better results than simply the median.

**Other Specific Comments**

page 1, line 20: are based on the solar signals; line 25: you cite examples with French and German radar(s); I hope you do not mind to cite here the Swiss network, too (although, our analysis, is simply based on the median, I know, as stated in several other points … )

page 2 line 12: … which is based on the difference (on the Log-transformed, decibel scale) of the daily median H and V reflectivities  of the strongest 21 hits in each days [that means the $11^{th}$ value …]

page 3, I would remove Fig. 1; part of the corresponding space could be used to show the symmetric form of the MeteoFrance Trappes radar in color (while in Holleman et al. 2010b, Fig. 2, page 883, it is in black and white and consequently, more difficult to be read and interpreted); furthermore, the reader would have these nice and informative pictures for two radars in the same paper. (Maybe you have it of other radars? Detailed co-polar and cross-polar info regarding radar antennas are rare in peer reviewed literature and one has often to look for info in technical reports of PhD dissertations, e.g. Reimann (2013)

page 4,5,6, from Eq.( 2) to (15): could/would the authors reduce the number of Equations, for instance by citing (and not writing again) the ones that have already been published?

Page 5, Table 1: why do you list radar KUO (never used) and you do not list radar UTA? (which is used in Fig.3); please, substitute KUO with UTA.

Page, Fig. 3: To what day does it refer? Could you provide examples for ANJ and LUO, too?

Page 8, line 5-6, "… is in addition affected by the differences in the Tx power between the channels". Are you referring to the magic-Tee ? If so, I find difficult to imagine the power in not split properly; I would write " … could be in addiction affected by …

Page 8, line 8-9, "… because …to their normal operation"; are you referring to the T/R Limiter?

Page 9-10: as stated, I would make it considerably shorten and delete Fig. 4 and 5.

Page 12, figure 7, two bottom pictures: I would set y-axis with the same limit for both ELEV and AZIMUTH, e.g. from 0.9° to 1.3° in order to better emphasize different (apparent) widths because of azimuth scanning (PPI instead of RHI)

By the way, Page 13, text commenting figure 7: do not you think that the effective beamwidth in azimuth is larger as a consequence of the operational azimuthal scan? (I mean, PPI). I think it is worth commenting this effect and also citing Zrnic and Doviak (1976).

**Suggested references:**

Zrnic, D.S. and R.J. Doviak, 1976: Effective antenna pattern of scanning radars, *IEEE Trans. Aer. Electronic Systems,* **12**, 551-555

Reimann J., 2013: On Fast, Polarimetric Non-Reciprocal Calibration and Multi-polarization Measurements on Weather Radars, *PhD dissertation*, University of Chemintz, ISSN 1434-8454, DLR Forschungsbericht 2013-36, 161 pages

---

## Author Comment (AC1) · 13 May 2016

Authors' response on the review of manuscript: amt-2016-52

Improved Analysis of Solar Signals for Differential Reflectivity Monitoring

Asko Huuskonen, Mikko Kurri and Iwan Holleman

We thank the referees for a careful reading of the manuscript and for the comments and recommendations given. The comments have helped us to correct errors in the original manuscript and also have helped us to improve the clarity of the presentation in several places.

Our detailed responses are included after each comment in blue.

Review of "Improved analysis of solar signals…" by A. Huuskonen et al. 2016.

The authors describe procedures for obtaining the system ZDR bias and quint angle using solar radiation. The paper contains valuable data and its results can be used for the monitoring of ZDR calibration in operational weather radars. To make the authors' findings clearer, I'd recommend considering the following issues.

1. It was not clear to me how ZDR of the solar flux has been obtained in the Finnish radars. In the calculations of ZDR, the IRIS subtracts the system ZDR from ZDR measured in all range gates (see also page 15 lines 6, 7) including the gates with solar hits. Let denote the ZDR biases introduced by the three major radar components, i.e., the transmitter, receiver and antenna, as $B_T$,  $B_R$, and $B_A$ correspondingly. The authors consider the antenna bias as a component of the receive bias, i.e., $\Delta R_{dr} = B_R + B_A$ in eq. (17). It is not clear from the manuscript whether the authors consider the antenna bias as a part of transmitter bias as well, i.e.,   $\Delta T_{dr}$ = $B_T + B_A$. I assume this here. The system ZDR ($ZDR_{sys}$) is

$ZDR_{sys} = \Delta R_{dr} + \Delta T_{dr} = B_R + B_A + B_T + B_A = B_R + 2B_A + B_T.$                    (A)

This is eq. (17) rewritten in more detail. Now consider the measurements in the solar flux. The radar processor subtracts the system ZDR from measured ZDR values. ZDR from the sun is $B_R + B_A$ before subtracting $ZDR_{sys}$. So the reported ZDR from the sun ($ZDR_{sun}$) is:

$ZDR_{sun} = B_R + B_A - ZDR_{sys} = -B_A - B_T = -\Delta T_{dr}$        (B)

Eq. (B) shows that the reported ZDR depends on the bias in transmit. $ZDR_{sun}$ depends on the bias in receive as well. Thus the reported solar ZDR depends not only on the bias in receive as the authors stated throughout the text but on the transmit bias as well. If $B_R$ changes and $ZDR_{sys}$ have not been yet adjusted by rain measurements, then $ZDR_{sun}$ changes as well. So the reported $ZDR_{sun}$ depends on the receiver and transmitter biases. Please clarify if this is correct.

Indeed, when system bias is subtracted from solar Zdr, we see the transmitter bias negated, assuming that the system bias is set correctly. Yet it is correct to say that solar method is monitoring the receiver path only . If something changes in the transmitter chain, excluding the antenna, no effect is seen in the solar Zdr (unless the system bias is changed). On the other hand, if something changes in the receiver chain, the solar Zdr changes readily. This also makes physical sense as the solar signal is independent of the transmitter. This was also noted by Holleman et al (2010, p. 885).    We have clarified this so that in Fig. 8 we present the solar Zdr without the system bias subtraction. We discuss the system bias subtraction in 4.3, after presenting the zenith scan data.

2. The ZDR scatterplot from the solar spikes shown in Fig 2 (the right panel) exhibits quite strong diagonal disturbances.    Such a feature has not been observed in French, German, and US radars. To make this feature more pronounced, a scatterplot from a distinct solar scan is desirable. Could you please show data from a box solar scan when the antenna scans the solar disk. Such data can be obtained with the IRIS routines by setting up a sector scan with an angular step of 0.2 deg . Such data could already exist in the radar data archive.

A "saddle" ZDR scatterplot makes it difficult to match it with a parabolic surface eq. (8).    The ZDR surface is a difference of two parabolic surfaces eq.(2), i.e., it should be a parabolic surface as well.    The observed ZDR saddle surface raises a question of its origin. I wonder if this is a feature of the antenna. Please compare    the ZDR diagonal disturbances with the placement of the antenna struts that support the feedhorn.    Are there 4 antenna struts placed about 45 deg to the horizon?

The difference of two parabolic surfaces is either elliptic or hyperboloid, depending on whether the curvatures are equal or opposite in azimuth and elevation. Widths in Table 1 and the curvature equations tell that the curvatures are of opposite signs, in agreement with Fig 2 (right panel). In a saddle surface there are directions where the values of the function are equal to that at the origin. When the curvatures are closely equal this happens in diagonal directions, just as seen in Figure 2. The four struts supporting the feed are also in the diagonal directions.    The antenna beam is formed by the antenna disk, the feed, the struts and the

radome etc.    We cannot tell which the origins of the form we observe are! A sector scan on sun is enclosed below. The quantity presented is 10*log10(Z_V/Z_H) and hence the signs are opposite to those in Figure 2.

3. It is recommended in the manuscript to obtain the system ZDR by subtracting the fitted powers from the sun in H and V channels. The radar reports $Z_H$ and ZDR from which $Z_V$ is obtained as $Z_V = Z_H – ZDR$ (eq. (12) and also p.7 line 8). It is stated (p.7 line 8) that this implies $C_H = C_V$. The latter means that the system ZDR is zero dB, which can be not the case for radar. A nonzero system ZDR implies that $C_H$ differs from $C_V$ because amplifications and losses in the polarization channels are different. So it is not clear from the manuscript how the system ZDR bias affects the calculation of $Z_V$.

We acknowledge that the assumption Ch=Cv is erratic. We wish to thank the referee for pointing this out. We have reformulates the section based on this comments and the comments by referee #2.

It is being recommended obtaining the ZDR bias by subtracting the fitted solar powers in the channels (page 12). Then what is the purpose of section 3.3 where the modeled ZDR signature is considered?

The formulation is very useful when explaining the form of the Zdr saddle surface in Figure 2. Also some of the methods compared and shown in Figure 6 are based on direct fitting to Zdr.

4. Measurements in rain with a vertically pointed antenna could be affected by a water film on the radome. A rain film on a radome is not ideally uniform that leads to different attenuation of H and V waves. For low antenna elevations, ZDR can be distorted by more than 1 dB at a wet radome (e.g., https://ams.confex.com/ams/96Annual/webprogram/Paper288057.html ) Similar effect could be expected at vertical incidence. So I wonder if ZDR from rain at vertical incidence is so perfect.

Indeed a wet radome can produce Zdr distortion at low elevations. At the top of the radome the rain film is thinner and but may not be uniform. We can safely assume that a 360 degree scan reduces the distortion, but some distortion might remain. We have not made any comprehensive studies of the issue but we have not noticed that the results in overhanging rain would be any different as compared to the cases where the rain reaches the ground. We have added a sentence at the beginning of section 3.4 about reasons of using a 360 degree scan.

It follows from Fig. 3 that rain has been observed at altitudes as high as 8 km. These are very high altitudes for Finland. The radar volumes at such heights are above the melting layer, most likely. Radar UTA in the figure is absent in Table 1.

In most cases even in the summer the precipitation is solid at the top parts of the profile. Climatologically the melting layer is at 3 km altitude in Finland, and in the cases presented the melting layer top was below that altitude.    We have never noticed any melting layer effects in the results, based on years of zenith profile data. We have replaced "rain" with "precipitation" in several places throughout the paper, and pointed out that the profile extends above the melting layer. We have corrected Table 1, where UTA is indeed absent and KUO is included, even though not used in the paper.

After transmitting a radar pulse, radar receivers can be out of their normal stage during a time interval equivalent a range of 8 km (page 8 line 10). Most likely, no rain can be present at this height. How do you calibrate ZDR in such situations?

If a radar returns to the normal operation only at 8 km, the number of successful calibrations is lower. Especially at winter time calibration measurements are hardly obtained. Yet it is possible to calibrate those radars during the summer. Most of our radars stabilize at much shorter ranges, and even the slowest one actually at 7 km. This can be handled with changing the limiter. It is also beneficial for the reflectivity measurements that the response is fast. For the Zdr case it is actually sufficient that the H and V channels have closely identical response curves.

Signals with SNR > 5 dB have been used in the analysis (page 8 line 13). Has noise been subtracted from the measured powers? What is SNR of the sun flux and how noise is processed in the solar hits?

Noise has been subtracted of the measured powers. The solar peak SNR is typically 7-8 dB (see Fig. 2), and the noise is subtracted just as for all data.

Figure: LDH and ZH measured by the IRIS Suncal utility.

[Figure]

Review of manuscript: amt-2016-52

Title: Improved Analysis of Solar Signals for Differential Reflectivity Monitoring Authors: Asko Huuskonen, Mikko Kurri and Iwan Holleman

GENERAL COMMENTS

This manuscript reviews and improves several aspects of the methodology that uses solar hits found in operational scans for daily monitoring of the calibration of the weather radar differential reflectivity (ZDR) bias in reception. The main achievements are the proposal of a procedure to improve the accuracy of the power and differential reflectivity estimated for the solar hits and that the authors provide solid arguments supporting a separate fit to solar hits' power in each channel, in contrast to a direct fit to solar ZDR data. The authors also give good guidelines about how to interpret the solar ZDR bias results in comparison to zenith ZDR measurements in rain in order to identify potential inconsistencies in the transmitter or gain losses. I believe that this work is of interest for the weather radar community and fits well within the scope of AMT journal. Comments, questions and suggestions to be considered by the authors are listed in the following..

SPECIFIC COMMENTS

- Eq. (1) given in P.3 is valid when the calibration reflectivities $dBZ^{H,V}$ are available for both channels. In this case both radar constants CH and CV are known -note that the radar constants CH and CV include antenna parameters and transmission losses regarding their respective polarisations. However, often the radar processor only allows for a single calibration reflectivity and the differential reflectivity offset, $dBZ^{H}$ and ZDR0 (e.g ), for instance. In this second case, a single radar constant is used for the two channels (e.g. CH ) and the differences between radar constants are accounted for in the ZDR0 calibration. If the measurands available are ZH and Zdr and only CH is known (the difference CH − CV is accounted in the calibration ZDR0), PH can be calculated from Eq. (1) but it is not possible to calculate PV . Instead the following quantity can be calculated:

    PV − (CH − CV )= PH − Zdr = ZH − Zdr − CH − f(r)

Therefore, Eq.(12) also holds valid for this case and it is not correct to say that CH = CV (P.7, L.8). In this case, separate fits would mean fitting PH and PV − (CH − CV ) but the difference between the peak powers would still give an estimate of the differential receiver bias.

I think that these two cases need to be more explained and the procedure to follow in the two fitting approaches (direct Zdr fit and separate H/V fit) explicitly described for each case.

The FMI system is of the second type, where horizontal reflectivity and differential reflectivity are available and single radar constant (CH) is used for both channels. Hence the formulation in the comment is valid for our case, corrects our error, and improves the clarity of the derivation. We have rewritten section 3.3 as suggested later so that we first present direct Zh, Zv fitting, then Zh, Zdr fitting and finally present how the results are affected when only one radar constant is available.

- If I am not mistaken, the Zdr value corresponding to perfect antenna-sun alignment estimated from the fit has a slightly different interpretation depending on whether ZH and ZV are separately available with their own calibration reflectivities (Frech, 2013) or whether ZH and Zdr are available (Holleman et al., 2010a) together with the calibration $dBZ^{H}$ and ZDR0. In the first case, $\hat{Z}$dr carries information about the differential sensitivity (i.e. differences in noise figure between channels) while in the second case Zdr gives information (except for the ZDR0 calibration value) about the bias of the linear depolarisation ratio or differential receiver bias. I have tried to prove this in the Appendix at the end of this document.

The FMI system of the second type (ZH+Zdr) hence discussed in our paper. The result in Appendix is in agreement with statement in Holleman et al (2010a); the deduced quantity is the offset of the linear depolarization ratio. It is interesting to note that the interpretation of the result if the first case (ZH+ZV) is different. We think that discussion of such systems is beyond the scope of our paper.

- P.7, Eqs.(15)-(16): Since the squint angle is studied in an upcoming section, it is important to mention that these equations are derived assuming that the pointing biases are equal for both polarisations. Also Eqs.(10)-(11); the parameters $B_{\varphi,\theta}$ give the position of the minimum/maximum/saddle point of the surface but represent the biases only under the aforementioned assumption.

We need not actually assume equal pointing biases in the derivation, which is more evident in the new derivation of the equations. But our conclusion is indeed the one suggested by the referee; the position of the maximum/minimum/saddle point represent the bias only if the pointings of H and V agree.

We have removed "squint angle" from the text. Even though the squint angle has been used in some recent documents to mean the angular difference of the H and V pointing, it is mostly used to denote the angle between the antenna bore sight and beam directions. There is a great risk of misunderstanding, which is avoided by not using the term at all.

- P.6-7, Section 3.3: I suggest that this section is reordered for clarity and rigour; it should start with Eq.(12) to arrive at Eq.(8), since the latter model is the result of subtracting the two paraboloid surfaces. Then, Equation sets (9)-(11) and (13)-(16) could be joined in a single set. Having Eqs.(13)-(14) before the sentence in lines 18-19 would also be helpful. In addition, the first paragraph of the section may be moved after the mathematical derivation so that is easier to understand the qualitative description of the left panel in Figure 2.

The suggested structure is very good, and improves the section. We have rewritten the section accordingly.

- P.8, Fig.4: The median estimate (red) and the mean estimate (green) lines overlap but their corresponding window upper and lower widths are not at the same distance from each other. There seems to be an inconsistence in the green lines drawn for the mean.

We have corrected an error in the drawing routine which caused this. Now "median" and "mean" values are plotted correctly, in the original version both were plotted at the "median" value.

- P.9, L.8-9: I could not find the solar hit rejection explained in sentence "In addition solar hits with standard deviation ..." in the references given at the beginning of the paragraph. Do you apply it for the present work only? If so, maybe it should be explained or this sentence should be placed later in the section. Also, what is approximately the standard deviation expected for a solar hit?

Removing data with high standard deviation is a rather standard procedure used in data analysis. We have reformulated the text and moved the discussion on the applied limit to a later section.

- P.9, L.11-12: Just a "picky" comment: the method of removing outliers using the fit curve as reference may fail also when there is a single outlier but "badly located" so that it alone biases the fit curve. This is implied in the sentence just afterwards "The results are further improved ..." but may be interesting to mention it explicitly.

This type of situation does not arise when working with systems where the antenna pointing is close to correct. But the situation might arise and is worth mentioning. Hence we have added a note to the text.

- P.10, L.26: For clarity, this sentence may be extended to explain that, after the estimation of the width and the filtering, the mean value for ranges > 50 km is computed.

We have clarified the text as suggested

- P.10, L.28-30: Another reason to use a fixed width may be that at low elevations there might be too few (or none, depending on the maximum range) range bins at high altitudes available for estimation of the filtering window width.

In that case there would most probably also be too few range bins to estimate the solar power! Hence this is not a realistic case. No change made.

- P.10, L.30: Could you give a value for the fixed window width recommended for filtering (e.g. the one estimated using data at high altitudes in the analysis in Fig. 4)?

Typical value is between 2 and 2.5 dB, depending on the system. The width of the solar hit distribution depends on the number of samples used in the averaging. Typical values for the standard deviation are 0.6-0.8 dB in our case. The width recommended above amounts to three standard deviations which contain more than 99 % of the solar hit distribution. We have added a recommendation to the text.

- P.9-10, Section 4.1: I think it would be interesting to provide further evidence on how the proposed

filtering increases the number and quality of the solar hits, to support what is stated in P.2 L.15-16. For example, are the fit estimates more stable or the RMSE of the fit lower when applying the proposed quality control? This is somewhat accomplished in P.14, L.2-6 but comparisons of the number of hits and standard deviation for the same dataset before and after the quality control may be desirable.

There is an improvement between using the mean filtering ("blue") or the two other methods. In the case presented in the paper, the number of solar hits increases by 10 % and the RMSE decreases by 3 %. The improvement varies from case to case and depends especially on the number of rainy days in the dataset. We have added a short text to describe the improvement and also point out that the improvement is case dependent.

• P.11, L.7-9: I think the 3-parameter fit with fixed pointing may give biased results if there is a significant squint angle (from the reasoning in the comment above for P.7, Eqs.(15)-(16)).

Indeed that would happen. See next comment.

• • P.11, L.12: In this sentence it is not clear which methods are compared. The first method described (the one used in Holleman et al. (2010a)) is not analysed. Maybe this should be explicitly mentioned in case the reader expects to find it among the methods compared in Fig. 6.

We have decided to remove the 3-parameter fit with fixed pointing from the methods in Figure 6, because using the H-pointing for Zdr is not safe if the H and V pointings do not agree. Instead we have included the original Holleman et al (2010a) method.

• P.12, L.5-6: The sentence "For surfaces ... is one of the fit parameters" is a very good point and hints to a very strong argument to support the separate fit for the horizontal and vertical channels. Even if it is further discussed in the conclusions, I think it would be beneficial to put more stress in this reasoning at this point of the manuscript. The case described (constant surface) corresponds to an extreme case of ill-conditioning of the inverse problem in Eq.(8) (there are more parameters in the model than needed to explain the observations). This example also indicates that fitting Eq.(8) is not an appropriate methodology for routine usage, since the well conditioning of the problem depends on the magnitude of the differences between the horizontal and vertical widths, which may change in time and from radar to radar.

These are the very reasons why separate fits are recommended and it is appropriate to stress that more in the text. We have modified the text accordingly.

• P.11-14, Section 4.2: Could you specify which measurands were available for the calculations in this section?

We used horizontal reflectivity and the differential reflectivity. We have added a description of   the measurands in the text.

• P.15, L.16-17: If I am getting it right, assumed that the antenna gains are correct and that the bias corresponds to inaccurate transmitter losses, then this bias is systematic for the reflectivity values and should not affect the estimation of the pointing errors or the squint angle.

The text on P15 deals with the zenith calibration and hence is not dealing with the pointing. In the solar method the statement is true. The pointing is statistically independent of the power and width values. Hence any systematic error on the power has no effect on the pointing, and neither on the width actually.

TECHNICAL CORRECTIONS

We thank the referee for the detailed comments on the text and will take those into account when preparing the final manuscript. In addition we have corrected an error related to the scanning strategy of the FMI radars. In the present schedule 12 elevation angles are scanned instead of 11 angles in 5 minutes.

The authors should make clear the limitations of the bias calculations from the sun scan measurements. From the sun one can establish the overall gain bias, but the antenna effects although included are not equivalent to the antenna effects that are present in the two way measurements. There are some key assumptions that underpin bias measurements with the sun scan. These are not articulated in the paper. Here I list the constraints.

 (text not shown)

Going back to the paper I examined the values from table 1. So I compute the $G_h$-$G_v$ + 10*log[H(el,H) H(az,E)]-10*log[ V(el,E) V(az,H)];   that is ($g_hB_{h1}$/ $g_vB_{v1}$) in dB scale and obtained:   -0.045,   -0.03,   -0.182, 0.12,   -0.015 for the five radars respectively. Thus I conclude that for the radars No 1, 2, and 5 the sun scan could provide bias to within ☐ 0.1 dB.

The difference of the one-way pattern (solar Zdr), and two-way pattern (zenith Zdr) may indeed limit the accuracy of the comparison of the two data sets. We think that when evaluation integrals in Eq.3 one should take into account the diameter of the radio sun (0.57 degrees), which corresponds to -1.5 dB point of the one-way antenna pattern. Hence only the center part of the antenna pattern counts here. As $G_h$ and $G_v$ are known, the final result actually tells that the uncertainty due to the antenna mismatch is at most only 0.05 dB for the radars. This is probably also an upper limit for the result if the sun size is taken into account. The main use of the solar Zdr is in the monitoring of the stability of the receiver chain. For that use time-invariant uncertainties such as this have no significance. As the effect seems to be rather small, we have not added any discussion of the effect in the paper.   Including it would require a number of new equations to be derived and presented, extending the scope of our paper unnecessarily.

Some other comments:

If the radar is scanning the sun the position is wider in azimuth because the effective beamwidth is larger.

The larger azimuth width is indeed explained by the fact that the antenna is moving in azimuth during the scanning. The actual increase depends also on the windowing function which is applied to the data in the signal processing. It is possible that the windowing affects the width so that the actual width in azimuth is not much larger than the width in elevation. For example, Huuskonen et al (2014a) show image where the azimuth width is only 1.1 times larger than the width in elevation. They also provide a thorough discussion on the issue. In the present case a rectangular averaging window is in use, and hence the azimuth widening due to antenna movement is well visible. We have added an explanatory sentence to the text.

You state "uncorrected".   It is clearer to indicate that "Ground clutter had not been applied to the data, and noise power has not (or has?) been removed."   Uncorrected is confusing and strictly would mean that you had correct data and then you "uncorrected it".

"Uncorrected" means that ground clutter filtering has not been applied but that noise power has been removed. This is a good point, although we think that the reference to the Doppler filtering clarified the case. We have reformulated the sentence for clarity and kept "uncorrected" which is widely used in this context.

Page 14, line 5:   Remove "because"    Removed
Page 16, line 5: "one months of data"   Corrected

Fig. 5 not clear – Caption states "Probability" whereas on the axis I see "Power fraction".

The label has been modified accordingly

Caption Fig. 2 end sentence you repeat the.

Indeed, "the" is repeated in the caption of Fig. 4.

Sentece on page 5 "In addition solar hits with standard deviation much larger than that expected for a solar hit (e.g. three times larger) have been rejected." Is this really what you are doing? If so how are you computing the standard deviation, local running average in azimuth at a constant altitude, or else? Or, do you really mean the data that deviate by thee standard deviations from the mean are discarded?

The standard deviation is calculated from the solar hit profile which is obtained by applying Eg.1 We then have estimates of solar hits along the radial, from which we calculate the mean and can also calculate the standard deviation. This sentence has been modified based on the comment by referee #2

**A review of "IMPROVED ANALYSIS OF SOLAR SIGNALS FOR DIFFERENTIAL REFLECTIVITY MONITORING" by Asko H. et al., submitted to AMT (amt-2016-52)**

The paper describes a number of analysis methods for monitoring the residual differential bias between horizontal and vertical reflectivities of solar signals. Personally, I find it interesting and stimulating. I also believe that is of interest not only for the academic world, but also for the operational meteorological radar community, since dual-polarization receivers are now available in several Countries. The paper shows nice, intuitive and informative picture such as Fig. 2, 3, 6, 8 and 9. Maybe it could be made a bit shorter and more concise? I am confident that by doing so, it will attract more readers. Aiming at this, I would suggest, for instance, to shorten in particular Sec. 4.1, by deleting also Figures 4 and 5. Finally, it is probably worthwhile warning the readers about the (relatively weak) intensity of the Solar(+Noise) Signal: ~10 dB SNR with high-gain and high-sensitivity radar as long as the beam axis is hitting the center of the solar disk; but this is not the case for most of the hits acquired during operational monitoring …

Several other specific comments and suggestions to be considered by the authors are listed below.

Locarno Monti, 31.3.2016

Yours faithfully, Marco Gabella

**Fundamental, interesting, "hot" topics [from my (biased!) viewpoint]**

Page 3 Eq. (1), term $-2ar$ , which is two-way gaseous attenuation: is the term $a$ in your Radar Signal Processor set to a constant values, independently of altitude? Curiosity: what value   of $a$ (in dB/km) at the sea level do you use?

A constant value of 0.008 dB/km is being used in the signal processor. The same value is used when calculating the solar powers.

Page 7, just after Eq. (16); I would re-write the sentence in a completely different way; I mean, your great idea of using solar hits during operational scan for monitoring weather radar performances, has been indeed useful for many national weather services … Hence, I would go along that tradition by changing the sentence into a recommendation. For instance: unfortunately some radar software systems do not provide both H and V reflectivities, but instead only the H reflectivity and the differential one. In this way, when deriving Zv from the two previously-mentioned quantity, one has to assume that CH=Cv, which is not necessarily always the case. Or something similar; with this kind of formulation, future dual-polarimetric radar users will always deal with $Z_H$, $Z_V$ and $Z_{DR}$ and not only with two of them; there are other advantages associated to this choice: for instance, $Z_{DR}$ can be set to 0 dB by introducing an appropriate offset, while leaving $Z_H$ and $Z_V$ untouched and calibrated at their best according to the Probert Jones Equation (nominal values of Tx, Rx Losses and antenna Gain).

As pointed out by referee #2, the assumption of Ch=Cv is not correct. We noticed that we can do the same analysis when Z_H and Zdr are available than when both H and V reflectivity exist. Hence systems with Z_H and Zdr can provide sufficient information for the solar analysis, but indeed many things are simpler if both H and V available fully calibrated. We have added a recommendation but not as strict as suggested noting the fact the solar analysis is possible even without having both.

Page 8: paradoxically, you just briefly mention the important issue of Signal-to-Noise ratio in Sec. 3.4 ("Calibration of $Z_{dr}$ during rain"), which deals with "RELATIVELY STRONG" hydrometeor signals and you do not mention it when dealing with "much WEAKER" solar signals. On the one hand, please, consider a 10 dBZ echo say at 10 km altitude (when vertically pointing): SNR is of the order of ~20 dB for the Finnish high sensitivity radars. On the other hand, even when the antenna beam axis is hitting the center of the solar disk, the Sun+Noise "signal" is just ~10 dB stronger than the Noise; even worse, during the operational scan, the beam axis rarely hits the center of the Sun, so that typical SNR is probably around ~8 dB or less … I ask to authors to mention this important aspect and warn the readers about it: maybe you could anticipate it in the introductory part? Another important point: here at MeteoSwiss, ZH and ZV values are derived after Noise subtraction, so

that solar flux estimates "back-retrieved" from reflectivity values in the PPI are "intrinsically noise subtracted". Probably, the same is at FMI and it is worthwhile to inform the readers about it.

For FMI radars the solar SNR is somewhat below 10 dB, as shown in Fig2. Hence the solar signal is seen much beyond the 3 dB point, making a 5-par fit possible. The situation is different if the SNR is lower, for instance only 3 dB. Then the solar analysis becomes much more limited in capacity. The FMI solar estimates are noise subtracted just as for the MeteoSwiss radars. We have added text on low SNR cases in section 4.2 and recommend using 3-par fitting instead.

Eq (17) and Fig. 6 (linked with Fig. 8 and page 15, see my next comment): I would plot data only from Aug. 1 to Aug 20, before the discontinuity that is only subsequently described at page 15, line7-8. Alternatively, I would anticipate here such sentence, maybe even in the caption. More important: am I missing something? By simply looking at the picture, I would conclude that Zh(5par fit)-Zv(5par fit) [CYAN curve] is BIASED of approximately +0.1 dB with respect to the median (GREEN curve), which in turns look unbiased. Why? Curiosity: what is the st. dev. of these first 20 points? (it looks of the order of 0.05 dB or less…) Adding up, sorry to say, from Fig. 6, I do not conclude that 5-param-fit is better than median. Why do you conclude that the median is not recommendable? Finally, do you have explanation for the mean having a negative BIAS with respect to the median?

We consider it good to present this kind of case here to show that the solar Zdr results includes the system Zdr bias, and that solar Zdr changes whenever the system bias is changed. The actual value of the solar Zdr (e.g. being close to zero) is not a measure of its quality. We assess the various methods by their ability to estimate the value of the Zdr at the direct pointing to the sun. This is obtained by the fitting methods. The fact that the median or the mean are biased with respect to the fitting results is decisive when determining if they are recommendable methods. We have described the reasoning more in detail in the text. The standard deviation of the first 20 points varies between 0.04 and 0.05, depending on the method, as suggested.

Page 14 (text related to Fig. 8); line 3: dB instead of degree;   Indeed

line 5: yes, MeteoSwiss figures (0.05 dB for Albis, 0.06 dB for Lema and Dole, Gabella et al. 2015) were obtained using both wet and dry days; however, as stated at page 51, Sec. 2, of Gabella et al. 2015, "In the MeteoSwiss approach, only the last 5 km (60 gates) of each radar bin are averaged for Sun power retrieval"; this means that our solar echoes are at stratospheric and mesospheric altitude … Probably here it is worthwhile to emphasize that MeteoSwiss figures have been obtained: A) by neglecting atmospheric attenuation (affecting in a different way the 15 different elevations used); B) by simply using the median, instead of Zh(5par fit)-Zv(5par fit).

Many factors affect the results just as described above. We have added a short descriptive text also for the MeteoSwiss results, without describing all factors which affect the results.

Fig. 8: could you please add μ±σ also for March data? (e.g. in the upper left part of each picture) Could you please add a picture for UTA radar?

The March data gives closely the same results. Hence we have changed the numbers to cover both months. We have added UTA as the 5th radar in the figure. We noticed an error in the calculation of the standard deviation, which is now corrected. The new std's are somewhat higher.

Fig. 9: could you please add μ±σ also for March data? (e.g. in the upper left part of each picture) Could you please add a picture for UTA radar?

We have changed Fig.9 similar to Fig.8. In addition we now display the Zdr bias itself. In the original figure the system ZDR bias had been removed from the data. We consider showing Zdr bias as more informative. This is emphasized be changing the y-label to "Zdr bias [dB]"

Page 15, line 12-13. This is indeed an important and interesting statement and a key point of the paper, I think; you could be less concise and try an interpretation of your results. In terms of reproducibility and repeatability, I am confident that μ±σ for March data will help the interpretation. (You could even add more months…) However, in order to make conclusions regarding this interesting point you should also provide a table with H and V Tx Losses and antenna Gain.

The March data is nearly identical to April data, as given above. The differences of the transmitter losses and antenna gains are given in Table 1. We have reformulated the text to make the conclusions clearer.

**Conclusion**, page 16, line 4-6. Here, I would list fact and figures in a more neutral and specific way: instead of "statistical accuracy … is better than 0.02 for most radars" I would write "the standard deviation of differential reflectivity for 30 daily samples (April 2015) is 0.02 dB for two radars, 0.04 and 0.05 dB for the other two radars" or something similar … [will you add also the 5th UTA radar, please?]. "This is significantly better of what reported earlier, namely 0.2 dB for the MeteoFrance Trappes radar [and then please specify 90 days; by the way, was it Zh(5par fit)-Zv(5par fit)? Maybe you can split the Trappes 90 days in three monthly subsets and tell us it st. Dev. is ~0.2 dB for all three subsets or whether it varies (I hope and I am confident it does not… ) Similarly, you can explicitly mention the DWD figure (Frech, 2013): what are the number of days and constraint used? … (5-par? Or median? atmospheric attenuation correction included?) and MeteoSwiss ones. As stated at the top of the page (no atmospheric correction, simply using the median, not the 5-par fit … ) The number of days used for the 3 MeteoSwiss radar was larger than two hundreds (see Table 4, Gabella et al. 2015).

The standard deviation of a time series does not depend on the length if the statistical properties of the series do not change. Obviously we would get somewhat different values if we divide a time series into several sections, because std itself is a random variable with a distribution, and systems may change with time. Also with increasing time series length the risk of outliers is increasing which then would increase the std estimated value. This could easily be circumvented by using a more robust estimator, such as median absolute deviation (MAD) as the width estimator.   We see that it is beyond the scope of our paper to try to explain in detail where certain std number originate from. We do not refer to any DWD values, because no std values are given in the paper. The original FMI std values were incorrect, and the new values are closer to those by   Gabella et al.

Finally, I think it is worth mentioning other examples; for instance, as reported in Gabella, M.; Huuskonen, A.; Sartori, M.; Holleman, I.; Boscacci, M. The solar slowly varying component as detected by dual-and single-polarization C-band weather radars in Europe, Electromagnetic Wave Propagation and Scattering in microwave Remote Sensing, Technische Universität Chemnitz Publikationen, submitted on September 29, 2015
during a common observation period of 100 days in 2014, the dispersion of the difference between H and V, namely the standard deviation of 100 values of 10 Log(H/V), was 0.06 dB for Albis and **0.08 dB for ANJ** (see Table 3 in the above-mentioned paper). While using the **median** instead of **5-par fit** for **ANJ**, it was **0.12 dB** ; also this other example shows that using 5-par-fit gives better results than simply the median.

The method described by Gabella et al uses differences of the median H and V powers (median(H)-median(V), actually difference of the 11th largest H and V powers) , which is different to the median of the differential reflectivity (median(H-V)) described in our paper. Hence the result is not directly comparable to our findings. The observation that 5-par fitting gives better results than a difference of single power values makes sense.

**Other Specific Comments**
page 1, line 20: are based on the solar signals;

We have changed the text as "microwave signals from the sun"

line 25: you cite examples with French and German radar(s); I hope you do not mind to cite here the Swiss network, too (although, our analysis, is simply based on the median, I know, as stated in several other points … )

Indeed. We have added the Gabella (2015) paper, which was missing from the list.

page 2 line 12: … which is based on the difference (on the Log-transformed, decibel scale) of the daily median H and V reflectivities of the strongest 21 hits in each days [that means the $11_{th}$ value …]

We have changed the text based on the proposal.

page 3, I would remove Fig. 1; part of the corresponding space could be used to show the symmetric form of the MeteoFrance Trappes radar in color (while in Holleman et al. 2010b, Fig. 2, page 883, it is in black and white and consequently, more difficult to be read and interpreted); furthermore, the reader would have these nice and informative pictures for two radars in the same paper. (Maybe you have it of other radars? Detailed co-polar and cross-polar info regarding radar antennas are rare in peer reviewed literature and one has often to look for info in technical reports of PhD dissertations, e.g. Reimann (2013)

We rather keep Fig. 1 which is not published earlier anywhere. We do not discuss the Trappes radar in our paper; for our purposes a reference to the B/W is sufficient. Otherwise we agree that it would be good to have more of the images published.

page 4,5,6, from Eq.( 2) to (15): could/would the authors reduce the number of Equations, for instance by citing (and not writing again) the ones that have already been published?

It is at times difficult to find a right balance in presenting prior knowledge. We have rewritten most of section 3.3 and hence the equations (8) to (16) are largely new. Repeating Eqs.(2)-(7) is justified so that a new reader need not refer to the earlier literature to understand the derivation.

Page 5, Table 1: why do you list radar KUO (never used) and you do not list radar UTA? (which is used in Fig.3); please, substitute KUO with UTA.

We have an error in Table 1, which is corrected.

Page, Fig. 3: To what day does it refer? Could you provide examples for ANJ and LUO, too?

We have added a date and also mention the melting layer heights in text. Our aim is to show some typical examples and not the curves for the full network.

Page 8, line 5-6, "… is in addition affected by the differences in the Tx power between the channels". Are you referring to the magic-Tee ? If so, I find difficult to imagine the power in not split properly; I would write " … could be in addiction affected by …

We think our formulation is OK. Our intention is to point out that the power division affects the transmitter side only. Real systems are not ideal, and power mismatches exist.

Page 8, line 8-9, "… because …to their normal operation"; are you referring to the T/R Limiter?

Yes, we do. We now mention Limiter in the text.

Page 9-10: as stated, I would make it considerably shorten and delete Fig. 4 and 5.

We rather see this important and keep the text and the figures.

Page 12, figure 7, two bottom pictures: I would set y-axis with the same limit for both ELEV and AZIMUTH, e.g. from 0.9° to 1.3° in order to better emphasize different (apparent) widths because of azimuth scanning (PPI instead of RHI)

We indeed thought of this when preparing the figures. Having the same axis limits would show the difference between azimuth and elevation, but would affect the readability of the figure. Our aim is to point out the difference between H and V, and not between azimuth and elevation.

By the way, Page 13, text commenting figure 7: do not you think that the effective beamwidth in azimuth is larger as a consequence of the operational azimuthal scan? (I mean, PPI). I think it is worth commenting this effect and also citing Zrnic and Doviak (1976).

This is a well known effect, and discussed thoroughly in Huuskonen et al (2014a). In addition to the scanning in azimuth the actual width is affected by the windowing used in the signal processing. We have added a note on this, and also a reference to Zrnic and Doviak (1976).

Suggested references:
Zrnic, D.S. and R.J. Doviak, 1976: Effective antenna pattern of scanning radars, *IEEE Trans. Aer. Electronic Systems,* **12**, 551-555
 Reimann J., 2013: On Fast, Polarimetric Non-Reciprocal Calibration and Multi-polarization Measurements on Weather Radars, *PhD dissertation*, University of Chemintz, ISSN 1434-8454, DLR Forschungsbericht 2013-36, 161 pages